# Overview of normal behavior modeling approaches for SCADA-based wind turbine condition monitoring demonstrated on data from operational wind farms

Xavier Chesterman[1], Timothy Verstraeten[1], Pieter-Jan Daems[2], Ann Nowé[1], and Jan Helsen[2]

[1]Artificial Intelligence Lab, Vrije Universiteit Brussel, Pleinlaan 9, 3rd floor, 1050 Brussels, Belgium
[2]AVRG, Vrije Universiteit Brussel, Pleinlaan 3, 1050 Brussels, Belgium

**Correspondence:** Xavier Chesterman (xavier.chesterman@vub.be)

**Abstract.** Condition monitoring and failure prediction for wind turbines is currently a hot research topic. This follows from the fact that investments in the wind energy sector have increased dramatically due to the transition to renewable energy production. This paper reviews and implements several techniques from state-of-the-art research on condition monitoring for wind turbines using SCADA data and the Normal Behavior Modelling framework. The first part of the paper consists of an in-depth overview of the current state-of-the-art. In the second part, several techniques from the overview are implemented and compared using data (SCADA and failure data) from five operational wind farms. To this end, 6 demonstration experiments are designed. The first 5 experiments test different techniques for the modeling of normal behavior. The sixth experiment compares several techniques that can be used for identifying anomalous patterns in the prediction error. The selection of the tested techniques is driven by requirements from industrial partners, e.g. a limited amount of training data, and low training and maintenance costs of the models. The paper concludes with several directions for future work.

## 1 Introduction

In recent years, investments in renewable energy sources like wind and solar energy have increased significantly. This is the result of goals set in climate change agreements and changes in the geopolitical situation. According to the Global Wind Report 2022, an additional 93.6 GW of wind energy production capacity was installed in 2021. This brings the total to 837 GW, which corresponds to a 12% increase compared to the previous year (Lee and Zhao, 2022). To keep the transition on track, the profitability of the investments needs to be guaranteed. Furthermore, to keep the European economy competitive in the globalized market, the price of energy production using wind turbines needs to be kept as low as possible. Both depend to a large extent on the maintenance costs.

According to (Pfaffel et al., 2017), recent studies have shown that the operation and maintenance of wind turbines make up 25-40% of the levelized cost of energy. A more detailed analysis shows that premature failures due to excessive wear play a considerable role. These are caused by, among other things, high loads due to environmental conditions and aggressive control actions (Verstraeten et al., 2019), (Tazi et al., 2017), (Greco et al., 2013). If it would be possible to identify these types of failures well in advance, it would create the opportunity to avoid unexpected downtime and organize the maintenance of

turbines more optimally. This in turn would result in increased production and a further reduction of maintenance costs, which will improve the profitability of the investments and reduce wind energy prices.

This paper gives an overview of the current state-of-the-art on condition monitoring for wind farms using SCADA data and the Normal Behavior Modelling (NBM) framework. The focus on SCADA data is motivated by the fact that it is an inexpensive source of information that is readily available. This is valuable in an industrial context where adding new sensors is not straightforward and expensive. The focus on the NBM methodology can be justified by the fact that it has shown its merits, and that properly trained NBM models can result in interesting engineering insights. Several techniques used in the state-of-the-art research are also implemented and compared. For this, 10-minute SCADA data from five different wind farms is used. Furthermore, failure information is also available for these wind farms. More specifically, there is information on generator bearing, generator fan, and rotor brush high temperature failures. The different techniques are implemented and compared using six demonstration experiments. Five experiments focus on NBM and one focuses on the analysis of prediction error. By doing a comparative analysis and discussion of the results on real data, a better understanding can be achieved of the performance of different techniques on real data. Because an exhaustive overview is unfeasible, only a limited selection can be discussed. This selection is based on several assumptions, e.g. that there is only a relatively limited amount of training data and time, and that due to maintenance constraints the complexity of the methodology needs to be kept as low as possible. These constraints are based on feedback received from several industrial partners.

The paper is built up as follows. The first section is the introduction. The second section discusses the current state-of-the-art. In the third section, an experimental methodology is designed that combines, compares, and demonstrates the performance of several techniques mentioned in the state-of-the-art overview. The fourth part is the comparative analysis of several techniques from the state-of-the-art. The fifth and last part is the conclusion, which also includes a discussion of possible future directions for research.

## 2 Overview of the state-of-the-art

Failure prediction on wind turbines using SCADA data is a hot research topic. This is due to the fact that over time more sensor data has become available (Helsen, 2021). There are several different families of methodologies that compete in this domain. According to (Helbing and Ritter, 2018), the methodologies can be divided into model-based signal processing and data-driven methods. An alternative classification can be found in (Black et al., 2021), where a distinction is made between 1) trending, 2) clustering, 3) NBM, 4) damage modeling, 5) alarm assessment, and 6) performance monitoring. In (Tautz-Weinert and Watson, 2017) five categories are identified: 1) trending, 2) clustering, 3) NBM, 4) damage modeling, and 5) assessment of alarms and expert systems.

The NBM methodological family is very diverse. Many different algorithms can be used to model the "normal" or "healthy" behavior of a wind turbine signal. However, in all this diversity, there are several commonalities. Figure 1 gives an overview of a standard NBM flow. The SCADA data is ingested by the pipeline. The first step is splitting the data into a training and testing dataset. This is done prior to the preprocessing and modeling steps to avoid "information leakage". The testing dataset is a

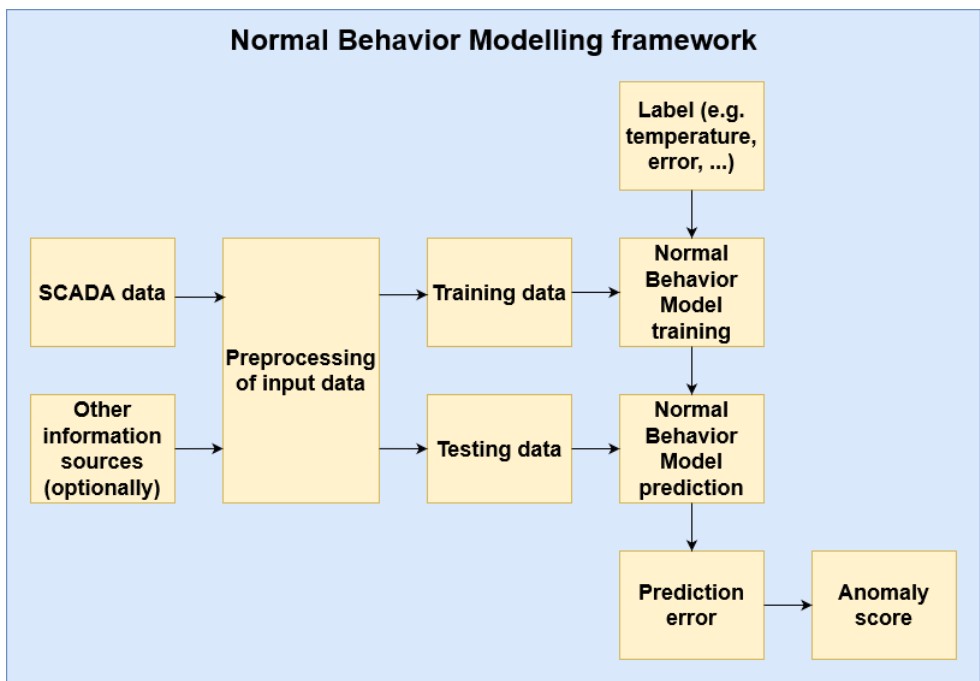

**Figure 1.** Schematic overview of NBM framework.

random subsample that is set aside for the final validation of the methodology and should not be used during training. How this split is made depends on the type and the amount of data. Often 80-20% or 70-30% random splits are made. However, other options are possible. If the data are time series, which is the case when using SCADA data, and the models used as NBM use
lagged predictors, then the train-test split should be done more carefully so that the relation between the target and the lagged predictors is not broken. A possibility is assigning the first 70% of the observations (based on their timestamp) to the training, and the rest to the testing dataset. In the next step, the data is preprocessed. This is done to clean the signals (e.g. removing measurement errors, filling in missing values, ...) and in some cases to reduce the noisiness of the signals (e.g. binning, ...). Filtering is sometimes used if it is expected that the relation between the signals is influenced by certain other factors (e.g.
wind turbine states, ...). The preprocessing is done on the training and testing dataset separately. However, the same techniques are used on both datasets.

    In the next step, the training dataset is used to learn or train the normal behavior model. For this, a health label of some kind is required. In the case of SCADA-based anomaly detection for wind turbines, this is in general a temperature signal that is related to the failure that needs to be detected. For example, if the research is focused on predicting generator bearing failures,
then the label might be the temperature of a generator bearing. This means that it is a supervised regression problem. Many different algorithms are suitable as NBM model, e.g. Ordinary Least Squares (OLS), Random Forest (RF), Support Vector Machine (SVM), Neural Network (NN), Long Short-Term Memory (LSTM), ... The NBM model is trained on healthy data (meaning not polluted with anomalies that can be associated with a failure). Once the NBM model has been trained it can

be used for predicting the expected normal behavior on the test dataset. In the next step, the difference between the predicted
and the observed behavior is analyzed. If there is a large deviation between the two, this can be considered evidence of a
problem. The deviation is in general transformed into an anomaly score that says something meaningful about for example the
probability of failure or the Remaining Useful Life (RUL).

In what follows an overview is given of different techniques that are used in the state-of-the-art literature for each step of the
pipeline. The papers that will be discussed in this section have the following properties: firstly, they are based on wind turbine
SCADA data, secondly, they perform condition monitoring and anomaly detection on temperature signals of the turbine (this
excludes for example research that focuses on the power curve), and thirdly, they follow the NBM methodology. By limiting the
scope of the overview, it can be more exhaustive, and give the reader a better insight into what has been tried in the literature.

## 2.1 Preprocessing techniques

Preprocessing is an important, although often somewhat underexposed, part of the NBM pipeline. Decisions taken during this
step can influence the training and performance of the NBM models later on. Different preprocessing techniques exist and have
been used in recent research. The choice of a technique is to a certain extent guided by the properties of the input data. E.g.
for time series data the order of the data points, and the relation between them, is relevant. This means that only preprocessing
techniques that retain this property of the data should be used. But even then multiple preprocessing techniques are usable.
Why a certain technique is chosen over a different one is often not thoroughly explained in papers. This subsection attempts to
give an overview of which techniques are used in current state-of-the-art research. An analysis of the literature shows that the
preprocessing of the data is used for among other things the handling of missing values, outliers, noise reduction, filtering, and
transforming the data.

Missing values can be problematic for certain statistical and machine learning models. For this reason, they need to be
treated/filled in properly. Several techniques are used in the literature. A first technique is removing the observations with
missing data (see (Maron et al., 2022), (Miele et al., 2022), (Cui et al., 2018) and (Bangalore et al., 2017)). This can be difficult
when time series modeling is used. Furthermore, the question needs to be asked why the data is missing. If it is not Missing
Completely Ad Random (MCAR) this can result in bias (Emmanuel et al., 2021). A different solution is single imputation. A
first example of this is carry forward and/or backward. In this technique, the missing value is replaced by the last known value
preceding the missing value (carry forward) or the first known value following on the missing value (carry backward). This is
used in (Bermúdez et al., 2022), (Campoverde et al., 2022), (Chesterman et al., 2022), (Chesterman et al., 2021) and (Mazidi
et al., 2017). An alternative is interpolation. This can be done in several ways, e.g. Hermite interpolation (see (Bermúdez et al.,
2022) and (Campoverde et al., 2022)), linear interpolation (see (Chesterman et al., 2022), (Chesterman et al., 2021) and (Miele
et al., 2022)), .... There are several elements that need to be taken into account when using these techniques. First of all, extra
care needs to be taken when using the previously mentioned techniques. Large gaps in time series are a problem since the
imputation can become meaningless. This can result in pollution of the relation between multiple signals. High dimensionality
of the data can also be a problem (Emmanuel et al., 2021).

If the amount of missing data is fairly limited (and it does not contain long stretches of missing data), an aggregate like the mean or median can be calculated and used as a proxy. The resulting time series has of course a lower resolution, but the missing values are gone. This method is used in (Verma et al., 2022). The success of this methodology of course depends on how much data is missing, and on how much data each aggregated value is based on. Another solution is what is called machine learning-based imputation. This technique uses certain machine learning models to impute the missing values (Emmanuel et al., 2021). There are several ways to do this. For example, clustering algorithms like k-nearest neighbors can be used to find similar complete observations. This is used in (Black et al., 2022).

The above overview gives several techniques that have been used in research that focuses on anomaly detection in temperature signals from wind turbines. However, there exist several other techniques that can be used for imputing missing values, e.g. hot-deck imputation, regression imputation, expectation-maximization, and multiple imputation (Emmanuel et al., 2021). To the best of the authors' knowledge, no (or a very limited number of) papers in this specific research domain have been written that use these techniques. The lack of research that uses, for example, multiple imputation (like Multivariate Imputation by Chained Equations or MICE (van Buren and Groothuis-Oudshoorn, 2011)) can be considered a blind spot.

A second problem that might influence the NBM training, is outliers. If these occur in sufficient quantity they can impact the modeling severely. Oftentimes the decision is made in the literature to simply remove them. This of course implies that the outliers can be detected in the first place. This can be done for example by using the interquartile range (see (Campoverde et al., 2022)), or a custom dynamic or user-defined threshold (see (Chesterman et al., 2022), (Chesterman et al., 2021) and (Castellani et al., 2021)), the 5-sigma rule (see (Miele et al., 2022)) or clustering (see (Cui et al., 2018) and (Bangalore et al., 2017)). Removing outliers needs to be done carefully to avoid that abnormal values associated with the failure of interest are also removed. This requires a good understanding of the data. In some cases the outliers will be clearly visible. This can for example be the case when measurement errors result in values that are multiple times larger or smaller than what is physically possible. In these cases removing the outliers is straightforward. However, in other cases, the difference between outliers or anomalies caused by the failing component and outliers caused by another reason is much less clear. Removing these types of outliers should only be done after careful consideration.

Some papers also use noise reduction techniques. Noise on signals can make it more difficult for the NBM algorithm to model the relation between them. If it is possible to clean the signal this should be considered, since it will improve the performance of the NBM model. This can be done for example by aggregating the data to a lower resolution (see (Chesterman et al., 2022), (Chesterman et al., 2021) and (Turnbull et al., 2021)), or cleaning or filtering the data using expert knowledge (see (Peter et al., 2022), (Takanashi et al., 2022), (Verma et al., 2022), (Turnbull et al., 2021), (Udo and Yar, 2021), (Beretta M. and J., 2020) and (Kusiak and Li, 2011)).

In some cases, it might be useful to transform the data. This can result in features with more favorable properties. For example, the Principal Component Analysis (PCA) transformation (see (Campoverde et al., 2022) and (Castellani et al., 2021)) and Zero-Phase Component transformation (see (Renström et al., 2020)) result in uncorrelated features which are linear combinations of the original signals. This can be beneficial for the training of the NBM. Another transformation that might be done is rebalancing the dataset. This can be necessary when certain operational states of the turbine are underrepresented in

the training data. Oversampling of the minority class or undersampling of the majority class is an option. A more sophisticated technique is the Synthetic Minority Oversampling Technique (SMOTE) used in for example (Verma et al., 2022). Lastly, in some papers, new features are created by clustering the original signals of the SCADA data in several groups using clustering algorithms. The new features are in the next step used as input to the NBM model (see (Liu et al., 2020)).

This overview shows that many different preprocessing techniques are available and have been tried. However, the technique choice is often not well motivated in the papers. Also, the impact of a certain technique on the results is in general not extensively discussed, even though it is known from statistical research that this can be significant.

## 2.2 The data and signals

SCADA data can come in different resolutions. The most available resolution is 10 minutes since it reduces the amount of data that needs to be transmitted (Yang et al., 2014). This means that the dataset contains for each 10-minute window the average signal value. In general, the SCADA data also contains information on the minimum, maximum, and standard deviation of the signal during the 10-minute window. Less common resolutions are for example 1-minute and 1-second. In the state-of-the-art literature, the following resolutions can be found:

- 10-minutes: (Bermúdez et al., 2022), (Black et al., 2022), (Campoverde et al., 2022), (Chesterman et al., 2022), (Maron et al., 2022), (Mazidi et al., 2017), (Miele et al., 2022), (Peter et al., 2022), (Takanashi et al., 2022), (Beretta et al., 2021), (Beretta M. and J., 2020), (Castellani et al., 2021), (Chen et al., 2021), (Chesterman et al., 2021), (Meyer, 2021), (Turnbull et al., 2021), (Udo and Yar, 2021), (Beretta M. and J., 2020), (Liu et al., 2020), (McKinnon et al., 2020), (Renström et al., 2020), (Zhao et al., 2018), (Bangalore et al., 2017), (Dienst and Beseler, 2016), (Bangalore and Tjernberg, 2015), (Bangalore and Tjernberg, 2014), (Schlechtingen and Santos, 2014), (Schlechtingen and Santos, 2012), (Zaher et al., 2009), (Garlick and Watson, 2009).

- 5-minutes: (Kusiak and Li, 2011).

- 10-seconds: (Kusiak and Verma, 2012).

- 1-second: (Sun et al., 2016), (Li et al., 2014).

- 100 Hz: (Verma et al., 2022), (Kim et al., 2011).

The 100 Hz data used in for example (Verma et al., 2022) comes from the Controls Advanced Research Turbine (CART) of the National Renewable Energy Laboratory (NREL). The fact that it is a research turbine makes it possible to sample at much higher rates than what is normally possible. Some papers combine the SCADA data with other information sources like event logs that contain wind turbine alarms (see (Miele et al., 2022), (Beretta M. and J., 2020), (Renström et al., 2020) and (Kusiak and Li, 2011)) or vibration data (see (Turnbull et al., 2021)).

The SCADA data contains information on many different parts of the turbines. This implies that the datasets consist in general of dozens or even hundreds of signals (depending on the turbine type). However, not all of them are relevant to

the case that is being solved. Some papers focus on a small subset of expert-selected signals (see for example (Peter et al., 2022), (Bermúdez et al., 2022) and (Chesterman et al., 2022), ...). Other papers use a large subset of signals and reduce the dimensionality of the problem during the preprocessing step or during a model-based automatic feature selection step that extracts the relevant information (see for example (Lima et al., 2020), (Renström et al., 2020), (Dienst and Beseler, 2016), ...). Some papers select signals based on the internal structure of the wind turbine (for example on subsystem level (Marti-Puig et al., 2021)). The advantage of the first method is that the number of signals used for training is limited, which reduces the computational burden of the training process. The disadvantage is however that for cases in which the expert knowledge is not complete, important signals might be missed. This is less likely when the second method is used. The disadvantage of this method is however that the computational cost is significantly higher and that the selected subset can change over different runs. The third method uses the wind turbine ontology or taxonomy as a guideline. Its performance depends of course on the quality of the ontology or taxonomy.

The signals that are often used for condition monitoring of wind turbines can be more or less divided into three groups: 1) environmental data like wind speed, outside temperature, ... (used in for example (Bermúdez et al., 2022), (Bermúdez et al., 2022), (Black et al., 2022), (Campoverde et al., 2022), (Mazidi et al., 2017), (Miele et al., 2022), ... ), 2) operational data from the wind turbine like active power, rotor speed, ... (used in for example (Bermúdez et al., 2022), (Black et al., 2022), (Chesterman et al., 2022), (Miele et al., 2022), (Peter et al., 2022), ...) and 3) wind turbine temperature signals like the temperatures of the generator bearings, the temperature of the main shaft bearing, the temperature of the generator stator, ... (used for example (Bermúdez et al., 2022),(Black et al., 2022), (Campoverde et al., 2022), (Chesterman et al., 2022), (Mazidi et al., 2017), ...). The first two groups are often used by default, independent of the target signal that needs to be modeled or the failure that needs to be detected. This is because they contain information on the wind turbine context (e.g. it is a stormy day, a very hot day, the turbine is derated, ...). The third group of signals is much more tied to the case at hand. For example, generator temperature signals are used if the focus lies on generator failures and gearbox temperature signals are used if gearbox failures need to be detected.

Overall it can be stated that most state-of-the-art research is focused on 10-minute SCADA data. This means that there are research opportunities on data with a higher resolution like 1 minute or 1 second. Furthermore, several signal or feature selection techniques are used in the literature. However, a thorough examination of their performance (expert knowledge based vs. automatic or model-driven vs. ontology or taxonomy-guided feature selection), advantages, and disadvantages, has according to the best of our knowledge not been done yet.

### 2.3 Normal behavior modeling algorithms

The next part of the NBM framework is the algorithm that is used for modeling the normal behavior. In general, a considerable amount of attention is paid in the literature to this. This part models the normal (or healthy) behavior of the signal of interest. For this, the current state-of-the-art literature uses techniques from the statistics and machine learning domains. These domains contain a large variety of algorithms that are suitable for the task. More or less three categories can be distinguished: 1) statistical models, 2) shallow (or traditional) machine learning models, and 3) deep learning models. Although this classification

gives the impression that the papers can be assigned to a single category, it is quite often the case that research uses or combines models from multiple categories.

Even though there are examples of recent papers in which statistical techniques are used for the modeling of the normal behavior, they are a minority. Models that are used are for example OLS and ARIMA (see (Chesterman et al., 2022) and (Chesterman et al., 2021)), where they are used to remove autocorrelation and the correlation with other signals from the target signal. A different kind of statistical algorithm that is occasionally used is the PCA (see (Campoverde et al., 2022)), and its non-linear variant (see (Kim et al., 2011)). These are, contrary to the previous algorithms unsupervised, but they can be used to learn the normal or healthy relations between the signals. In (Garlick and Watson, 2009) an OLS and an Autoregressive Exogenous (ARX) model are combined to model the normal behavior, which makes it possible to take time dependencies into account. The advantage of statistical models is that they are relatively simple, computationally lightweight, and data-efficient. They are well-studied, and their behavior is well-understood. This makes them often suitable as a first-analysis tool or in domains where there are constraints on the computational burden or the amount of data that is available. The downside is however that they are relatively simple, which makes them in general unsuitable to model highly complex non-linear dynamics. Whether this is a problem depends of course on the case that needs to be solved.

Techniques from the traditional (shallow) machine learning domain are used more often. With traditional machine learning is meant models like decision trees, random forests, gradient boosting, and support vector machines, ... These models are more complex than traditional statistical models and are better able to model non-linear dynamics. However, they require in general more training data and time. Examples of algorithms that are used in the current state-of-the-art are tree-based models like random forest (see (Chesterman et al., 2022), (Turnbull et al., 2021), (Kusiak and Li, 2011)) and gradient boosting (see (Chesterman et al., 2022), (Maron et al., 2022), (Beretta et al., 2021), (Udo and Yar, 2021), (Beretta M. and J., 2020), (Kusiak and Li, 2011)), and models like support vector machine and regression (see (Chesterman et al., 2022), (Castellani et al., 2021), (McKinnon et al., 2020), (Kusiak and Li, 2011)). Another type of model that is occasionally used is derived from the linear model (OLS) but includes some form of regularization to be better able to cope with high-dimensional data and highly correlated features, i.e. least absolute shrinkage and selection operator (LASSO) (see (Dienst and Beseler, 2016)). The latter model can be situated somewhere between the statistical and traditional machine learning categories.

In recent years deep learning models, e.g. neural networks, have become popular. Deep learning models are more complex than traditional machine learning models. The advantages are that they are even better at modeling non-linear dynamics. The disadvantages are that they require even more data, are computer-intensive to train, and the results are more difficult to interpret. This however has not diminished their popularity, and at the moment they are in the state-of-the-art research the most popular model category. Just like the traditional machine learning domain, the deep learning domain is very diverse. Over the years many different types of models have been developed. These are either completely new models or combinations of already existing deep learning models. Both can be found in the state-of-the-art literature. Deep neural networks are used in (Black et al., 2022), (Jamil et al., 2022) (transfer learning), (Mazidi et al., 2017), (Verma et al., 2022), (Meyer, 2021) (multi-target neural network), (Turnbull et al., 2021), (Cui et al., 2018), (Sun et al., 2016), (Bangalore and Tjernberg, 2015) (NARX), (Bangalore and Tjernberg, 2014), (Li et al., 2014), (Kusiak and Verma, 2012), (Kusiak and Li, 2011), (Zaher et al., 2009).

Another popular type of model is the Autoencoder (AE). Just like a PCA, this model learns normal or healthy behavior through dimension reduction which makes it ignore noise and anomalies. However, compared to the PCA it is better at learning non-linearities. This model type is used for example in (Miele et al., 2022), (Chen et al., 2021), (Beretta M. and J., 2020), (Beretta M. and J., 2020), (Renström et al., 2020), (Zhao et al., 2018). Convolutional Neural Networks (CNN), originally designed and used for the analysis of images, can also be used for the detection of anomalies and failures. Examples can be found in (Bermúdez et al., 2022) (combination of CNN and LSTM), (Xiang et al., 2022), (Zgraggen et al., 2021),(Liu et al., 2020). Another model that is used is the LSTM. This model is particularly suitable for time series since it is able to model the time dependencies. This model is used in (Bermúdez et al., 2022) and (Udo and Yar, 2021). Two other models that have also been used, but to a much lesser extent are the Extreme Learning Machines (ELM) (see (Marti-Puig et al., 2021)) and the Generative Adversarial Network (GAN) (see (Peng et al., 2021)).

Another type of algorithm that is occasionally used in the literature is based on Fuzzy Logic. An example of such a model is the Adaptive Neuro-Fuzzy Inference System (ANFIS). Papers that use this model are (Schlechtingen and Santos, 2014), (Schlechtingen et al., 2013) and (Schlechtingen and Santos, 2012). In (Tautz-Weinert and Watson, 2016) experiments are performed with among others ANFIS and Gaussian process regression. And finally, there is also research that uses copula-based modeling. An example is the research presented in (Zhongshan et al., 2018).

Overall it can be stated that the NBM ecosystem is diverse. In recent years, deep learning has become the most popular methodology. The merits of these models are clear from the results. Often they outperform the statistical and traditional machine learning models. However, the question is whether they are always the most suitable methodology for implementation in the field. The data requirements mean that deploying the system quickly on a new wind farm is not possible (transfer learning alleviates this issue to a certain extent). Also, the high computational requirements result in more costly retraining and higher maintenance costs. The question is whether these disadvantages weigh up against the improved performance once deployed in the field. Not much attention is paid in the literature to this question.

## 2.4 Algorithms for the analysis of the NBM prediction error

The last step of the NBM methodology is the analysis of the prediction error of the NBM model. This model predicts the expected normal behavior of a signal. If the true or observed signal deviates abnormally much from this prediction, or the deviation shows certain trends, then this might indicate that something is going wrong with the related component and that a failure is imminent. The main goal of the last step is to search for these patterns in the prediction error. There are many different techniques (and combinations of techniques) that can be used for this. There are different ways to classify them. They can be divided by domain. Firstly there are statistics-based methods that use the distribution of the prediction error under healthy conditions to determine a threshold that can be used to classify the prediction errors as normal or anomalous. Secondly, there are methods that are based on models from the Statistical Process Control (SPC) domain. Thirdly there are methods that are based on models from the machine learning domain. A different classification focuses on the number of signals they analyze in a single pass. There are univariate methods, which only take a single signal at a time into consideration. There are also multivariate methods, which look at multiple signals. In general, machine learning-based methods are multivariate. The

SPC-based method can be both since the univariate control charts algorithms like Shewhart, CUSUM, and EWMA have their multivariate counterparts. However, in the state-of-the-art literature often only the univariate versions are used. The statistics-based methods that use the distribution of the prediction error are in general univariate. An exception to this is the Mahalanobis distance which is mostly used in a multivariate setting.

The overview of the state-of-the-art given in this paper will use the first classification as a guideline. The technique that uses the distribution of the prediction error to find a suitable threshold to identify anomalies is for example used in (Meyer, 2021), (Zhao et al., 2018), (Kusiak and Verma, 2012), ... The Mahalanobis distance combined with an anomaly threshold is used in (Miele et al., 2022), (Renström et al., 2020), (Cui et al., 2018) and (Bangalore et al., 2017). SPC techniques are used in (Udo and Yar, 2021) (Shewhart control chart), (Chesterman et al., 2022) (CUSUM), (Chesterman et al., 2021) (CUSUM), (Bermúdez et al., 2022) (EWMA), (Campoverde et al., 2022) (EWMA), (Xiang et al., 2022) (EWMA) and (Renström et al., 2020) (EWMA). The machine learning-based methods for the analysis of the prediction error are in general modifications of traditional machine learning algorithms. For example, the Isolation Forest, which is used in (Beretta et al., 2021), (Beretta M. and J., 2020) and (McKinnon et al., 2020), is similar to the Random Forest algorithm, while the One-Class SVM, used in (Turnbull et al., 2021), (Beretta M. and J., 2020) and (McKinnon et al., 2020), is similar to the SVM.

Overall it can be stated that in the current state-of-the-art multiple techniques are used for the analysis of the prediction error, without a single category clearly having the upper hand. Furthermore, both univariate and multivariate techniques are still used. The multivariate techniques can analyze the prediction errors of multiple signals, which gives them an advantage compared to the univariate techniques. However, their disadvantage is that when analyzing several signals at the same time, a deviation in a single signal might be masked. This is shown in (Renström et al., 2020), where the authors observe that when the Mahalanobis distance is calculated on several prediction errors at the same time (multivariate setting), it does not always clearly increase when only a single prediction error deviates. For this reason, they point out that it would be interesting to combine multivariate and univariate techniques.

## 3 Design of demonstration experiments for evaluating the performance of techniques used in the state-of-the-art literature

In this section, the methodology of the experiments is explained, which will be used to demonstrate certain techniques found in the state-of-the-art literature. For this demonstration, an NBM pipeline is designed, which consists of the following steps: data preprocessing, NBM modeling, anomaly detection, and health score calculation. The pipeline is validated on data from five operational wind farms. The data contains information on 3 types of failures, e.g. generator bearing replacements, generator fan replacements, and rotor brush high temperature failures. In each step of the pipeline, multiple techniques and configurations are tested and compared. To this end, 6 experiments are designed. Care is taken to create as much as possible a lab environment. This means that the parts of the pipelines that are not relevant to the experiment are kept constant.

## 3.1 The input data

The experiments are based on two data sources (for confidentiality reasons not all the details of the input data can be shared). The first one is 10-minute SCADA data originating from 5 different onshore wind farms (wind farms 1-5). The geographic location of each wind farm is different. The wind turbines in these wind farms are all of the same type with a rated power of 2MW. The wind farms are relatively small, containing only 4-6 wind turbines. Some datasets contain a substantial amount of missing values. The number of (obvious) measurement errors is low. The SCADA data contains over 100 signals. Only the 10-minute averages of the signals related to the drive train or the operational condition of the wind turbine are used in this research. The signals that are selected from the SCADA data are based on the state-of-the-art literature. It is a relatively large subset, larger than what would be used if the selection is based on only expert knowledge. Table 1 gives an overview.

| Signal name | Symbol | Unit | Info |
|---|---|---|---|
| TempGenBearing_1 (avg) | $T_{gen\_bear\_1}$ | °C | Temperature of the first generator bearing. |
| TempGenBearing_2 (avg) | $T_{gen\_bear\_2}$ | °C | Temperature of the second generator bearing. |
| TempStatorWind (avg) | $T_{stator}$ | °C | Temperature of the generator stator. |
| GeneratorSpeed (avg) | $V_{gen}$ | RPM | Rotational speed of the generator. |
| Generator torque (avg) | $\tau_{generator}$ | Nm | Torque at the generator. |
| TempConvInlet (avg) | $T_{conv\_inlet}$ | °C | Temperature of the converter inlet. |
| TempGearbBear_1 (avg) | $T_{gear\_bear\_1}$ | °C | Temperature of the gearbox bearing 1. |
| TempGearbBear_2 (avg) | $T_{gear\_bear\_2}$ | °C | Temperature of the gearbox bearing 2. |
| TempGearbInlet (avg) | $T_{gear\_inlet}$ | °C | Temperature of the gearbox inlet. |
| GearboxSpeed (avg) | $V_{gearbox}$ | RPM | Rotational speed of the gearbox. |
| TempRotorBearing (avg) | $T_{rotor\_bear}$ | °C | Temperature of the rotor bearing. |
| Rotor speed (avg) | $V_{rotor}$ | RPM | Rotational speed of the rotor of the wind turbine. |
| Active power (avg) | $P_{active}$ | kW | Amount of power that is being produced by the wind turbine. |
| Nacelle temperature (avg) | $T_{nacelle}$ | °C | Temperature measured inside the nacelle of the wind turbine. |
| Outside temperature (avg) | $T_{ambient}$ | °C | Ambient temperature. |
| Wind speed | $V_{wind}$ | m/s | Wind speed measured at each turbine. |

**Table 1.** Overview of the SCADA signals used as input for the NBM model

The second source of information are the maintenance logs, which contain the replacements and failures. Information is available on three types of events: generator bearing replacements, generator fan replacements, and rotor brush high temperature failures. For each replacement or failure event, information is available on the turbine ID, the date, and the event type. This is valuable information for validating the methodologies. However using these logs has also a couple of challenges, e.g. imprecise event dates or missing events. The logs give only an approximate indication of when something went wrong. Furthermore,

it needs to be pointed out that a replacement of a component does not necessarily mean that the component had failed. Some components will have failed, others will have been replaced as a preventive measure.

## 3.2 Preprocessing

The first step of the NBM framework or pipeline is the preprocessing of the data (see Figure 1). Because the main focus of this paper lies on the NBM and the anomaly detection techniques, no extensive comparative analysis of different preprocessing techniques is done. However, the NBM pipeline makes use of several preprocessing techniques, which makes it necessary to at least discuss or mention them. Some techniques are trivial but necessary. They will just be mentioned without going into more detail. The more interesting ones, like for example the healthy data selection or the fleet median normalization, will be discussed more thoroughly.

The NBM framework makes use of the following preprocessing steps:

– Data cleaning: selection of relevant variables and turbines, renaming of the variables, matching of the SCADA data with the replacement information, and linear interpolation or carry forward/backward of the missing values (similar to what was done in (Bermúdez et al., 2022), (Campoverde et al., 2022), (Mazidi et al., 2017), (Renström et al., 2020) but with linear interpolation). This step will not be discussed in more detail since it is a trivial transformation.

– Selecting healthy training data: a rule-based method will be discussed in depth.

– Determining the operating condition of the turbines: this is done using the IEC 61400-1-12 standard (Commission, 2022) as a guideline.

– Signal filtering using the wind farm median: this is an important transformation with a significant impact on the results. For this reason, this step is discussed in depth.

– The removal of sensor measurement errors is done in a fully automated way. A short discussion of this step is given.

– Aggregating to hour level: the purpose of this step is to reduce the amount of noise and the number of missing values in the data. Similar actions were taken for example in (Turnbull et al., 2021) and (Verma et al., 2022). This step is not always appropriate. This is for example the case for failures that form very fast over time, or for signals that exhibit damage patterns that are very short-lived like in vibration analysis. It is up to the data analyst to determine whether the advantages weigh up against the disadvantages. This preprocessing step will not be discussed in more detail, since it is a trivial technique.

### 3.2.1 Signal filtering using the fleet (wind farm) median

The wind turbine signals in the SCADA data are quite complex, meaning there are a lot of factors that influence them. This complexity makes it more difficult to model the normal behavior. This means that if a part of this complexity could be removed

it would simplify the problem, which normally should result in an improved modeling performance and an overall more data-efficient model. This can be accomplished by calculating the fleet median of a signal (e.g. temperature of generator bearing 1 at time t for turbines 1, 2, 3, 4, 5) and subtracting it from the wind turbine signals (e.g. temperature of generator bearing 1 at time t of turbine 1). This technique is also used in (Chesterman et al., 2022) and (Chesterman et al., 2021). The fleet median can be seen as an implicit normal behavior model. It is implicit because it does not require the selection of predictors or the training of a model. It models the normal behavior as long as 50% + 1 turbines are acting normally at any given time. By taking the median over a whole farm, it captures farm-wide effects, which are common to all the turbines, which is why it will be called the "common component". Elements modeled by this component are for example the wind speed, the wind direction, and the outside temperature, ... By subtracting the median from the wind turbine signal, these farm-wide effects are removed. What is left are turbine-specific effects, which will be called the "idiosyncratic component". Turbine-specific anomalies should only be visible in there.

In practice, this preprocessing step means that from each signal in Table 1 the fleet median is subtracted (e.g. $T_{rotor}$ - fleet median $T_{rotor}$, $T_{rear}$ - fleet median $T_{rear}$, ...). Figure 2 shows the results of the decomposition for the 5 turbines of wind farm 2. The top subplot shows the original generator bearing 1 temperature signals. The middle subplot shows the common components. From the plot, it is obvious that the common component captures the general (fleet or farm-wide) trend, while turbine-specific evolutions are ignored. The bottom subplot shows the idiosyncratic components. The power down of turbine 4 is (more) clearly visible in the idiosyncratic component than in the original signal. This indicates that the decomposition is successful. Figure 3 shows that the fleet median is useful for filtering out macro-level or fleet-wide effects like seasonal fluctuations from the data. The common component captures clearly the seasonal fluctuations in the nacelle temperature, which results in an idiosyncratic component that is free of them. This is beneficial because it means that seasonal fluctuations will not influence the false positive rate when less than one year of training data is used. Furthermore, the common component also captures the transient behavior like cool-downs that are common to all turbines in the fleet. This means that the idiosyncratic component is free from most transient behavior. What remains of transient behavior are cool-downs that are unique to the turbine. This means that they are caused for example by a turbine that is turned off for maintenance. These events are relatively rare, which makes that subtracting the fleet median from the signals has reduced the modeling complexity substantially.

How well the fleet median removes the macro-level effects depends of course on the quality of the fleet median. An issue that may arise, especially in small wind farms, is that a substantial amount of turbines are offline for maintenance. For example, it can be that 2 turbines are offline for maintenance in a wind farm with 4 turbines. This will of course have an impact on how representative the median is for the normal behavior. For this reason, a rule-based safeguard is added that under certain conditions will convert the fleet median to NaN. The rules are the following:

– if fleet size $< 5$ : No missing values at time t are allowed.

– if $5 \leq$ fleet size $< 10$ : At most 20% missing values at time t are allowed.

– if fleet size $\geq 10$ : At most 40% missing values at time t are allowed.

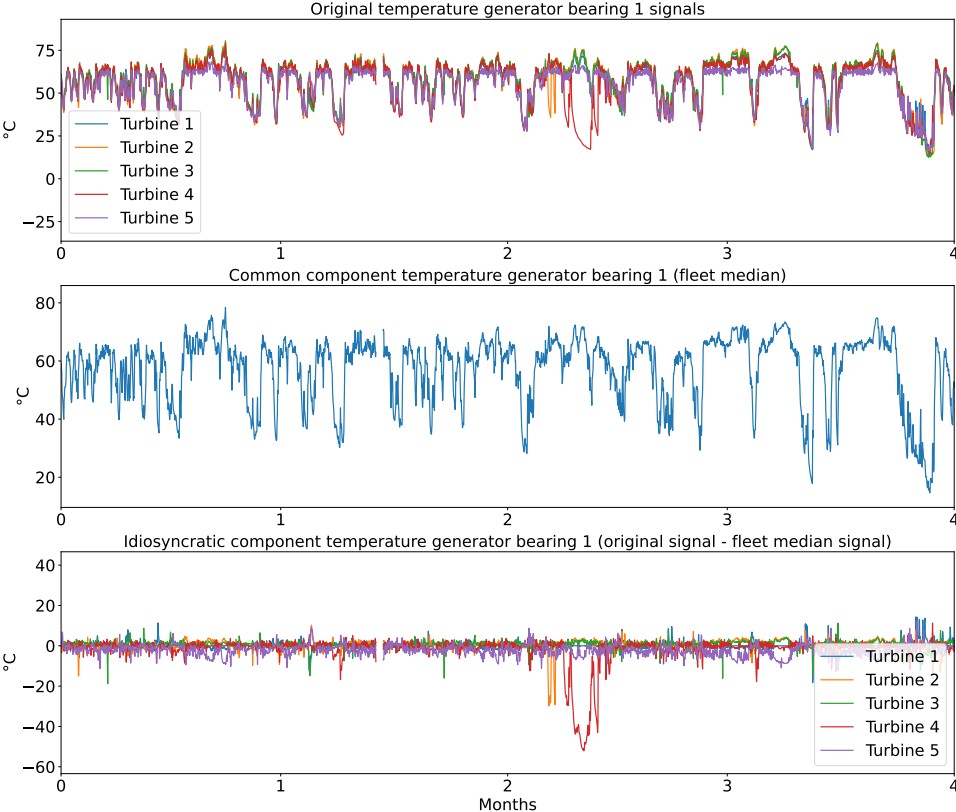

**Figure 2.** Example of the impact of decomposing the generator bearing 1 temperature of the turbines in wind farm 2 in a common and idiosyncratic component

### 3.2.2 Selecting healthy training data

The selection of healthy training data is an important step in the NBM framework. If it is not done properly, it can result in the contamination of the training data with anomalous or "unhealthy" observations, which can disturb the training of the normal behavior. Data is considered "healthy" if it is not polluted by abnormal behavior caused by a damaged component. Unfortunately, data from real machines does not contain a label that indicates whether it is healthy or not. This means that certain assumptions need to be made about the data, e.g. a "healthy data"-rule. The rule used in this paper considers data that precedes a failure by less than 4 months (which is the same as what is used in (Verma et al., 2022)) or follows on a failure by less than a month (to avoid test and upstart behavior) as unhealthy.

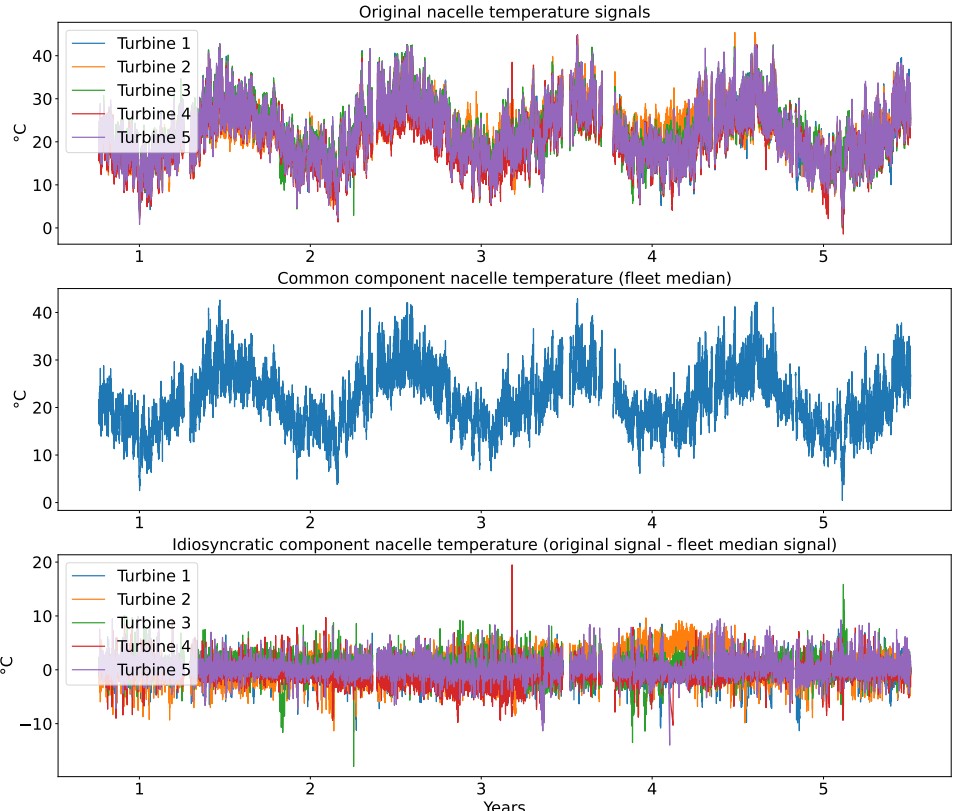

**Figure 3.** Example of the impact of decomposing the nacelle temperature of the turbines in wind farm 2 in a common and idiosyncratic component

Once the unhealthy data has been identified, healthy data can be selected. The methodology presented here selects the healthy data in a fully automated fashion. The user can determine how much training data (number of observations) per turbine is required for modeling the normal behavior. The healthy data is selected in chronological order from the time series. For most experiments, the first 4380 (which equals roughly 6 months of data) healthy samples of each turbine are selected for training. The selected data is combined into a single training dataset. This implies that only a single model per signal per farm is trained, which results in a large reduction of the training time, and more efficient usage of the training data. Figures 4 and 5 give a schematic representation of how the healthy training data is selected.

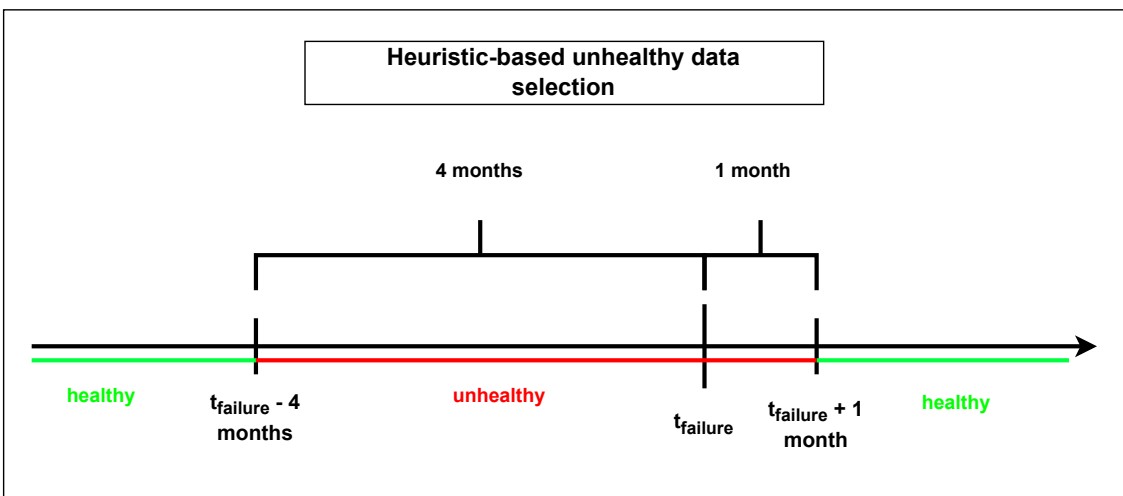

**Figure 4.** Unhealthy data identification

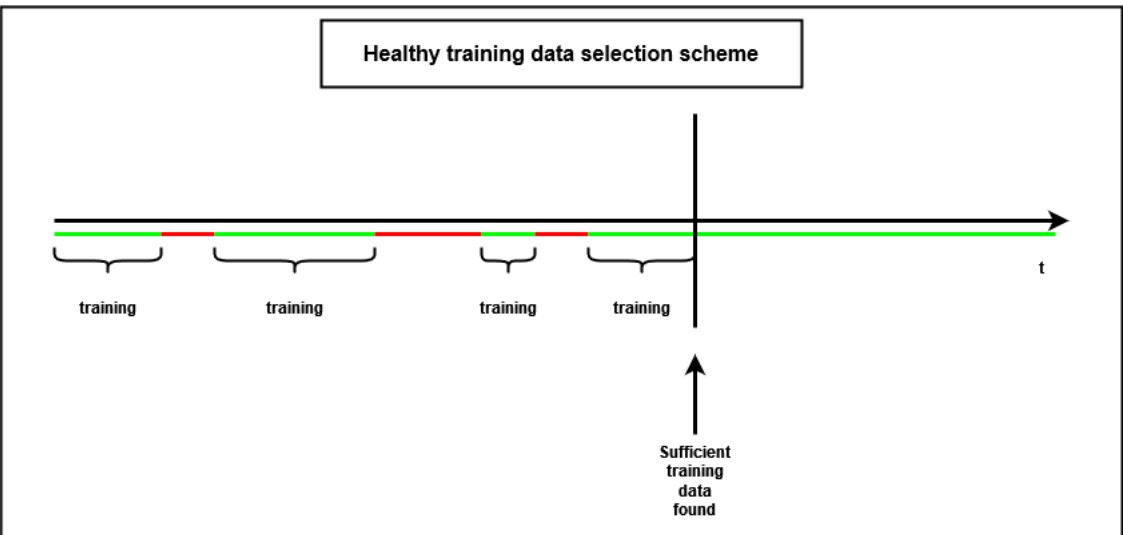

**Figure 5.** Heuristic-based healthy data identification

### 3.2.3 Handling of measurement or sensor errors

Removing outliers and/or measurement errors is something that is done in most research. The SCADA data used for the demonstration experiments contains a small amount of unrealistic values. The technique used here uses thresholds to determine which observations are measurement errors and which are not. The overview of the state-of-the-art shows that this is a technique that is often used. The thresholds used in the demonstration experiments are determined in a fully automated fashion so that they do not need to be manually determined for each signal of each turbine type. This manipulation of the data is not without

risk. Only measurement errors should be removed, not the deviations in the signals caused by the failure of the component. The threshold depends on the median value of the original signal (meaning that the fleet median has not been subtracted from the wind turbine signal). This value is multiplied with a scaling factor, which is in this paper set to 1. This threshold is used on the signal from which the fleet median has been subtracted. All absolute values in this signal that are larger than the threshold are considered measurement errors. Concretely this means that values from the wind turbine signal that deviate from the fleet median (on the positive side) by more than its median value are considered measurement errors. Eq. 1 shows the procedure using mathematical notation. The measurement errors are replaced with NaN values. In a later step, they are replaced using linear interpolation or if necessary carry forward/backward.

$$\text{Error}_{measurement} = \begin{cases} 1 & \text{if } |\text{Value}_{fleet\_corrected}| > \text{Value}_{original}\ \text{outlier\_factor} \\ 0 & \text{otherwise} \end{cases} \tag{1}$$

where:

$\text{Error}_{measurement}$ = Indicator whether the observation is a measurement error or not.

$\text{Value}_{fleet\_corrected}$ = Signal value from which the fleet median value has been subtracted.

$\text{Value}_{original}$ = Signal value.

outlier_factor = Constant multiplier.

### 3.3 Normal behaviour modelling

The normal behavior model is the core of the NBM framework. It models the "normal" or "healthy" relation between one or more predictors and a target signal. Once trained it can be used to predict the expected normal behavior. The differences between the observed and predicted values are then analyzed by anomaly detection algorithms in the final step. In this section, the methodological setup of the experiments will be discussed. To assess the performance of the NBM configurations, two metrics are used. The first metric is the healthy test data RMSE. A proper NBM model should be able to model the healthy data well. However, this metric does not tell much about how well it can distinguish unhealthy from healthy data. For this reason, a second metric is introduced, which is the difference between the median prediction error for the healthy and unhealthy data ($\Delta\text{PE}_{unh-h}$) (see Eq. 2). The idea behind this metric is that good NBM models should have a small prediction error on the healthy data ($\text{PE}_h$) (because they are trained on the healthy data) and a large prediction error on the unhealthy data ($\text{PE}_{unh}$) (because something has changed compared to the healthy situation). For the type of failures studied in this paper, $\Delta\text{PE}_{unh-h}$ should be positive because the damage to the related components should result in higher-than-normal temperatures. The more positive $\Delta\text{PE}_{unh-h}$ is, the better the NBM, because it indicates that the model can distinguish better healthy from unhealthy data.

$$\Delta\text{PE}_{unh-h} = \text{median}(\text{PE}_{unh}) - \text{median}(\text{PE}_h) \tag{2}$$

where:

$\Delta\text{PE}_{unh-h}$ = Difference between median unhealthy and healthy data prediction error.

$\text{PE}_{unh}$ = Unhealthy data prediction error.

$\text{PE}_h$ = Healthy data prediction error.

As a baseline the fleet median signal for the healthy training data is used. Since the fleet median also models the normal behavior, but without the requirement of specifying the predictors, it will be called an "implicit NBM" (vs. the "explicit NBMs" that do require a specification of predictors, model, parameters, ...). The explicit NBM that will mainly be used is the elastic net. It is a simple, transparent, and robust model that can handle large amounts of (correlated) predictors. At the same time, it can work with a limited amount of training data. This corresponds to requirements set by industrial partners, e.g. at most only a couple of months of 10-minute training data per turbine, low maintenance cost, low training cost, and high transparency. Nevertheless, in experiments 2 and 6 the performance of the elastic net will be compared to that of more complex models from the shallow machine learning domain (i.e. light gradient boosting machine (light GBM), support vector regression (SVR) in experiment 6) and the deep learning domain (i.e. multi-layer perceptron (MLP) in experiments 2 and 6). This should give an idea of the limits and usability of the elastic net model, and whether the trade-off between computational cost and complexity on the one hand and the performance on the other hand, is acceptable or not.

### 3.3.1 Elastic net regression for modeling the normal behavior

It has been shown in the literature that linear models can be good modelers of the normal behavior of wind turbines, and they are also time efficient (see (Dienst and Beseler, 2016)). However, by using elastic net (which was developed in (Zou and Hastie, 2005)), which is basically a linear regressor with $L_1$ and $L_2$ regularizers added to it (see Eq. 3), there are some extra advantages. Firstly, the model is more robust when many (correlated) features are used. Secondly, it also performs an automatic feature selection. This implies that it is possible to model for example the generator bearing 1 temperature by giving the model all the signals that are connected to the whole drive train of the turbine. This reduces the configuration burden for the user. Furthermore, the algorithm works in a transparent way, which avoids the "black box" problem. The amount of training data it requires is also favorable compared to more complex machine learning algorithms. A disadvantage of the model is that it is relatively simple, which means that it is not good at modeling highly non-linear dynamics. Whether this is a problem will be tested in experiment 5, where the performance of the elastic net is compared with that of more complex shallow machine learning and deep learning models.

$$\hat{\beta}^{elastic-net} = \underset{\beta}{\text{argmin}}\left\{\sum_{i=1}^{N}(y_i - \beta_0 - \sum_{j=1}^{p} x_{ij}\beta_j)^2 + \lambda_1 \sum_{j=1}^{p} |\beta_j| + \lambda_2 \sum_{j=1}^{p} \beta_j^2\right\} \tag{3}$$

where:

$\hat{\beta}^{elastic-net}$ = estimates of the coefficients or weights by the elastic net model.

$\beta_0, \beta_j$ = coefficients or weights of the model.

$\sum_{j=1}^{p} |\beta_j|$ = $L_1$ penalty term.

$\sum_{j=1}^{p} \beta_j^2$ = $L_2$ penalty term, also called Tikhonov regularization.

$\lambda_1 \geq 0$ = weight of $L_1$ penalty term.

$\lambda_2 \geq 0$ = weight of $L_2$ penalty term.

### 3.3.2 Training of the NBM model

The NBM models are trained on "healthy" data that is extracted from the SCADA data. Failing to train on more or less healthy data can result in severe degradation of the modeling performance of the NBM (this depends on the relative quantity of anomalies). To reduce the computational and maintenance burden of the pipeline, a single NBM model per signal per wind farm is trained. This means that healthy training data from several wind turbines are combined in a single training dataset. This decision was taken in response to concerns raised by wind turbine operators that if separate models would be trained per turbine this would result in an unacceptable maintenance burden. Combining training data from multiple wind turbines is however not without risks. Structural signal differences (e.g. a turbine with a generator bearing that is always 1 or 2 degrees warmer than the one of a different turbine under the same conditions) between the different wind turbines are not modeled (unless wind turbine dummies are added to the predictor list). This can result in structural deviations in the prediction errors, e.g. a prediction error that is structurally positive or negative. However, data analysis showed that the temperature or behavior differences between the different bearings are small. There are no indications in the results (see further) that the differences between the different bearings seriously hamper the analysis. Furthermore, experiments in which the model was retrained after each bearing replacement did not show any clear performance improvement.

In general, the training of the NBM models will be done by using the first 4380 healthy observations (or six months of data) of each turbine. This amount is limited on purpose so that it answers the requirements of the industry. Less training data means that new wind turbines can more easily be added to the anomaly detection system (less start-up time). The subtraction of the fleet median from the wind turbine signals neutralizes seasonal fluctuations. The NBM models are trained on the training data using a full grid search, or a random grid search when the number of hyperparameter combinations is large, over sensible ranges for the hyperparameters. To avoid overfitting 5-fold cross-validation is used. The trained model is tested on the test dataset to assess the performance. For each model, the healthy test data RMSE is calculated. This is used to compare the models from the different experiments. The third experiment examines the impact of further reducing the amount of training data to 2 months per turbine.

### 3.4 The anomaly detection procedure

The trained NBM model is used to predict the expected normal behavior. The prediction error of the model indicates how anomalous the observed behavior is. As shown in the state-of-the-art overview there are many different anomaly detection

techniques that can be used to analyze it. The techniques used in the sixth experiment are based on univariate statistical techniques that are transparent, robust, and computationally light. More specifically, two different techniques are tested. The first technique is based on the prediction error distribution. The second technique is based on a technique from the SPC domain.

The first technique is most suitable for identifying point anomalies. It is based on the principle of Iterative Outlier Detection (IOD) (also called Iterative Outlier Removal). This means that outliers are removed over several iterations until the outlier thresholds (these are the thresholds that determine which observations are outliers and which are not) have stabilized. To make these thresholds more robust for outliers, the standard deviation is approximated by the Median Absolute Deviation (MAD) (see Eq. 4 and 5, with k = 1.4826). The anomaly scores are calculated using Eq. 6.

$$\text{MAD} = \text{median}(|X_i - \tilde{X}|) \tag{4}$$

where:

$X_i$     = signal observation at time t = i.
$\tilde{X}$     = signal median.
MAD = median absolute deviation.

$$\hat{\sigma}_{robust} = \text{k MAD} \tag{5}$$

where:

$\hat{\sigma}_{robust}$ = robust estimate of standard deviation signal.
k       = constant multiplier or scaler.
MAD    = median absolute deviation.

$$\text{anomaly score} = \begin{cases} -3 & \text{if idio\_comp} > \text{median}_{idio\_comp} - 5\,\hat{\sigma}_{robust} \\ -2 & \text{if idio\_comp} > \text{median}_{idio\_comp} - 4\,\hat{\sigma}_{robust} \\ -1 & \text{if idio\_comp} > \text{median}_{idio\_comp} - 3\,\hat{\sigma}_{robust} \\ 1 & \text{if idio\_comp} > \text{median}_{idio\_comp} + 3\,\hat{\sigma}_{robust} \\ 2 & \text{if idio\_comp} > \text{median}_{idio\_comp} + 4\,\hat{\sigma}_{robust} \\ 3 & \text{if idio\_comp} > \text{median}_{idio\_comp} + 5\,\hat{\sigma}_{robust} \end{cases} \tag{6}$$

where:

idio\_comp            = idiosyncratic component.
$\text{median}_{idio\_comp}$ = median idiosyncratic component.
$\hat{\sigma}_{robust}$           = robust estimation of the standard deviation.

In the next step, the anomaly scores are transformed into health scores. This is done by calculating the moving average of the anomaly scores for different windows (1 day, 10 days, 30 days, 90 days, and 180 days). For these moving averages, upper and lower bounds are calculated. This is done by combining the moving averages with the same window length of the same signals from the different wind turbines and calculating Tukey's fences. Three positive thresholds are used to determine the moving average anomaly score. Next, the sum of the moving average anomaly scores is taken over the different windows for each timestep t. This sum is the health score and determines the health category (Eq. 8).

$$\text{MA}_{anomaly\ score\ win\ x} = \begin{cases} 1 & \text{if } \text{MA}_{winx} > \text{q}(\text{MA}_{winx}, 0.75) + 1.5\left(\text{q}(\text{MA}_{winx}, 0.75) - \text{q}(\text{MA}_{winx}, 0.25)\right) \\ 2 & \text{if } \text{MA}_{winx} > \text{q}(\text{MA}_{winx}, 0.75) + 2.5\left(\text{q}(\text{MA}_{winx}, 0.75) - \text{q}(\text{MA}_{winx}, 0.25)\right) \\ 3 & \text{if } \text{MA}_{winx} > \text{q}(\text{MA}_{winx}, 0.75) + 3.5\left(\text{q}(\text{MA}_{winx}, 0.75) - \text{q}(\text{MA}_{winx}, 0.25)\right) \end{cases} \tag{7}$$

where:

$\text{MA}_{winx}$ = moving average with window length x.

$\text{q}(\text{MA}, .)$ = . th quantile of moving average distribution.

$\text{MA}_{anomaly\ score\ win\ x}$ = moving average of anomaly score for window length x.

$$\begin{aligned} \text{health score } = &\ \text{MA}_{anomaly\ score\ win\ 1d} \\ &+ \text{MA}_{anomaly\ score\ win\ 10d} \\ &+ \text{MA}_{anomaly\ score\ win\ 30d} \\ &+ \text{MA}_{anomaly\ score\ win\ 90d} \\ &+ \text{MA}_{anomaly\ score\ win\ 180d} \end{aligned} \tag{8}$$

where:

$\text{MA}_{anomaly\ score\ win\ x}$ = moving average of anomaly score over window with length x.

$$\text{health category} = \begin{cases} \text{good} & \text{if health score} \leq 5 \\ \text{mediocre} & \text{if } 5 < \text{health score} \leq 10 \\ \text{bad} & \text{if health score} > 10 \end{cases} \tag{9}$$

The second technique is based on CUSUM (Page, 1955), which comes from the Statistical Process Control (SPC) domain. The CUSUM is designed to be more sensitive to small changes in the mean than for example the Shewhart charts (used for example in (Udo and Yar, 2021)). The algorithm is run with different subgroup sizes, e.g. 10 days, 30 days, 90 days, and 180 days. Instead of using the subgroup mean and the overall mean the subgroup median and overall median are used. The standard

deviation of the subgroups is replaced with the robust standard deviation estimated using the MAD (see Eq. 5). This makes the algorithm more robust against anomalous trends. For each subgroup size, anomaly thresholds are calculated using Eq. 7. For each signal, there are three thresholds. These are common for all turbines in the wind farm. The signal health scores are calculated by summing the anomaly scores for the different subgroup sizes (meaning for each time step t the sum is taken over all the subgroup sizes) (Eq. 8). The health category is calculated using Eq. 9.

Assessing the performance of the anomaly detection algorithms on real data is a non-trivial task due to data imperfections. Imprecisions in the replacement dates, problems that are not resolved after a first attempt, incomplete event lists, preventive maintenance, ... make it hard to automate the validation process. This means that each detection or non-detection needs to be validated by a human. Also, it introduces a certain inexactness in the validation process. For this reason, a somewhat different validation procedure will be used. The performance of the anomaly detection algorithms is assessed by calculating the percentage of failures that are correctly identified. This is the case when a cluster of bad health is found around the time of the failure. The ratio of false positives is also estimated. This is done using the following methodology. Firstly, turbines are selected that experienced no known failures. This is the case for 10 turbines in total. For those turbines, it can be assumed that the components were probably relatively healthy during the observation period. This means that the number of bad health observations will be fairly limited. Bad health observations that are found for those turbines are probably false positives. For each signal, the percentage of "bad" health observations is calculated. The median ratio for each signal over all the selected turbines is used as an approximation of the false positive ratio of the anomaly detection model.

## 3.5 The experiments

In total 6 demonstration experiments will be conducted. 5 experiments will focus on the NBM model, and 1 experiment will focus on the anomaly detection algorithms. Experiment 1 compares the performance of the base elastic net regression with that of the implicit NBM. Experiment 2 evaluates the added value of using lagged predictors. Lagged predictors have also been used in (Garlick and Watson, 2009). Experiment 3 analyzes the impact of reducing the amount of training data from 6 months per turbine to 2 months. In the state-of-the-art different amounts of training data are used. This is mainly driven by the amount of data available. Experiment 4 discusses the added value of PCA-transformed input for the elastic net. Using PCA for preprocessing of the data is also done in (Campoverde et al., 2022). Experiment 5 examines the added value of using more complex machine learning models like SVR (with a radial kernel) and light GBM. The performance of these models compared to that of the elastic net will say something about the importance of non-linearities. Experiment 6 compares the performance of the IOD-MAD and the CUSUM anomaly detection algorithms. For the analysis, the prediction error of the base elastic net model is used.

# 4 Results

## 4.1 Experiment 1: The added value of using the elastic net regression model on top of the results of the implicit NBM

Pipeline configuration: implicit NBM based on fleet median, explicit NBM based on elastic net regression, heuristic-based healthy data selection, full grid search hyperparameter tuning (5-fold CV), 6 months training data per turbine.

The first experiment investigates the usefulness of adding an explicit NBM (elastic net regression) model to the pipeline. One of the downsides of the implicit NBM (fleet median) is that it is unable to model turbine-specific transient behavior. Whether this is a serious problem depends on the case. However, if it is a problem it can be solved by adding an explicit NBM to the pipeline. If the above reasoning is correct, it can be expected that the healthy test data RMSE will decrease considerably if the elastic net regression is added to the pipeline.

Figures 6 and 7 show that the prediction error when using the elastic net, is indeed smaller than when only the implicit NBM is used. The most obvious improvement is that the large negative spikes in the prediction error of the implicit NBM, which correspond to cool-downs caused by power downs of the turbine, are much smaller in the prediction error made by the elastic net. This indicates that the elastic net is modeling the transient behavior to a certain extent. The error is however still larger during transient phases than during steady-state phases. The healthy test data RMSEs in Figure 8 further support the findings that the elastic net is a useful addition to the pipeline. The RMSEs of the prediction errors are substantially smaller when the elastic net is used.

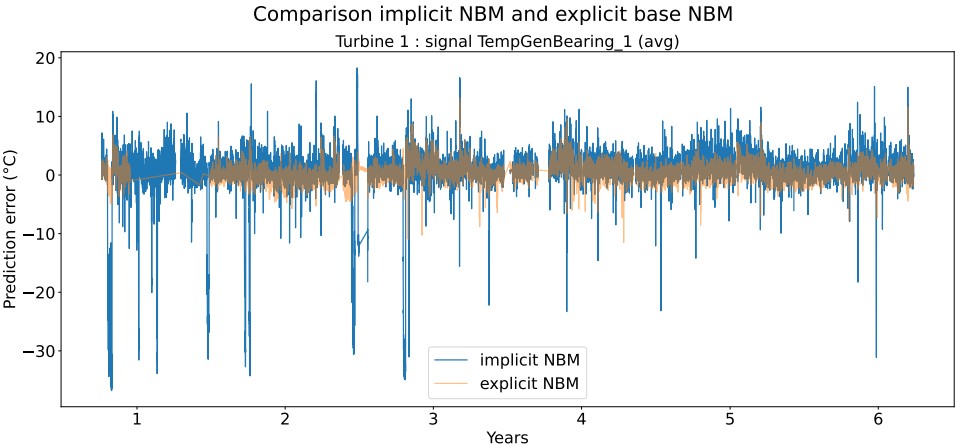

**Figure 6.** Prediction error explicit and implicit NBM model for the TempGenBearing_1 (avg) signal of turbine 1

Based on the results of the first experiment, it can be concluded that using the elastic net has a clear added value. The healthy test data RMSE is always smaller for the pipeline with the explicit NBM. The fact that the RMSE of the elastic net model is quite small, shows that relatively simple and lightweight models can be useful for the modeling of the normal behavior.

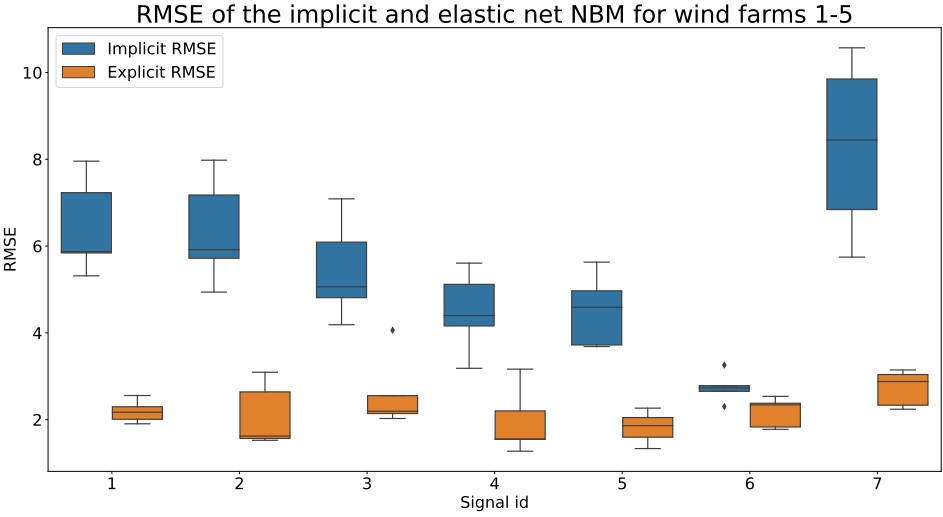

**Figure 7.** Prediction error explicit and implicit NBM model for the TempStatorWind (avg) signal of turbine 1

**Figure 8.** Comparison of RMSE of implicit and explicit NBM for wind farms 1-5. Signal ID 1 = TempGearbBear_1 (avg), signal ID 2 = TempGearbBear_2 (avg), signal ID 3 = TempGearbInlet (avg), signal ID 4 = TempGenBearing_1 (avg), signal ID 5 = TempGenBearing_2 (avg), signal ID 6 = TempRotorBearing (avg), signal ID 7 = TempStatorWind (avg).

## 4.2 Experiment 2: The added value of using lagged predictors

Pipeline configuration: implicit NBM based on fleet median, explicit NBM based on elastic net regression, heuristic-based healthy data selection, full grid search hyperparameter tuning (5-fold CV), 6 months training data per turbine, lags 1, 2, and 3 of each predictor are added.

In the second experiment, the input data is augmented by adding the lagged values of the input signals (excluding the target signal that is being modeled). The idea behind using the lagged terms is that it makes it possible to model the time dependencies. This can be useful when modeling the transient behavior of the turbine. In steady-state situations where factors like active power, ... change little, the positive impact will most likely be less clear. 3 lags (t, t-1, t-2, t-3) for each input signal are added. This means that the model can look up to 3 hours in the past.

Figures 9 and 10 show that the difference between the prediction errors for the NBM with no lags and the NBM with 3 lags is marginal. In general, there is no clear difference visible between the two. Surprisingly, there is also no clear improvement to be found in the modeling of the transient behavior. Figure 11 gives an overview of the RMSE on the healthy data for all the target signals for the turbines in wind farms 1-5. The results show that adding the 3 lags to the model results for a majority of the signals in a marginal reduction of the median healthy test data RMSE.

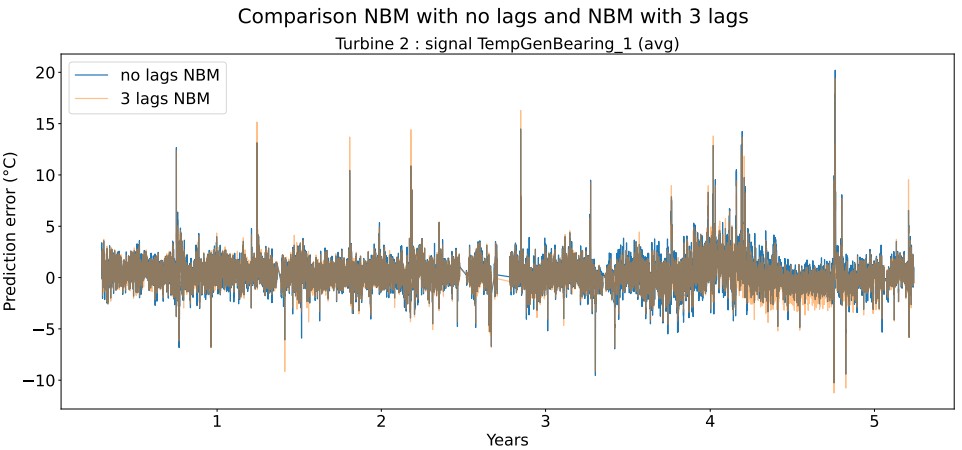

**Figure 9.** Prediction error explicit and implicit NBM model for the TempGenBear_1 (avg) signal of turbine 2

Figure 12 shows the difference between the unhealthy and healthy median prediction errors ($\Delta\mathrm{PE}_{unh-h}$ calculated using Eq. 2) for the elastic net and multi-layer perceptron (MLP). The figure focuses on the results for three target signals, i.e. TempGenBearing_1 (avg) ($T_{gen\_bear\_1}$), TempGenBearing_2 (avg) ($T_{gen\_bear\_2}$) and TempStatorWind (avg) ($T_{stator}$). For the three failures that are being examined, i.e. the rotor brush high temperature failure, the generator bearing failure and the generator fan failure, it is assumed that the degradation of the component can be observed directly or indirectly in respectively

the $T_{stator}$, the $T_{gen\_bear\_1}$ or $T_{gen\_bear\_2}$ and the $T_{stator}$, $T_{gen\_bear\_1}$, and $T_{gen\_bear\_2}$. More specifically, in all three cases, an increase in the temperatures is expected when the component is damaged. This means that $\Delta\mathrm{PE}_{unh-h}$ should be positive. The more positive it is the more useful the NBM is for anomaly detection.

    The results in Figure 12 show that for the elastic net the $\Delta\mathrm{PE}_{unh-h}$ only marginally increases when 3 predictor lags are used. For rotor brush high temperature failures $\Delta\mathrm{PE}_{unh-h}$ is clearly positive for the elastic net model with 0 or 3 lagged

predictors. This corresponds with the expectations. However, $\Delta\mathrm{PE}_{unh-h}$ is also for $T_{gen\_bear\_1}$ clearly positive. This is more difficult to explain since it is unlikely that the rotor brush high temperature failure can be linked to abnormal high temperatures

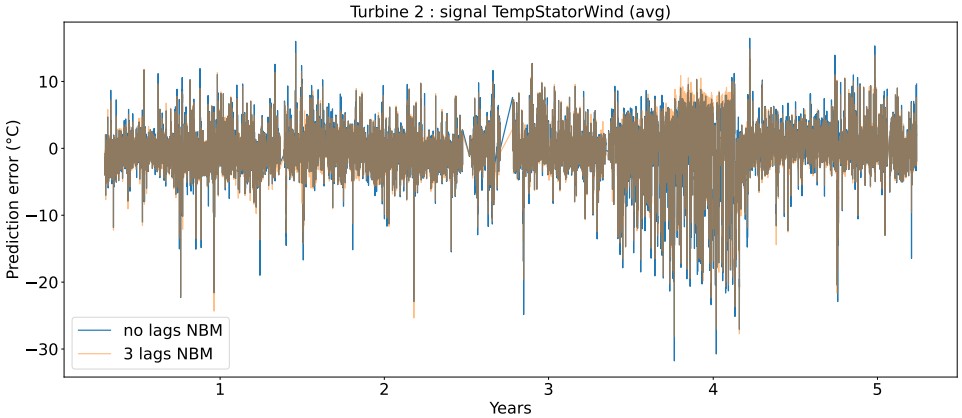

**Figure 10.** Prediction error explicit and implicit NBM model for the TempStatorWind (avg) signal of turbine 2

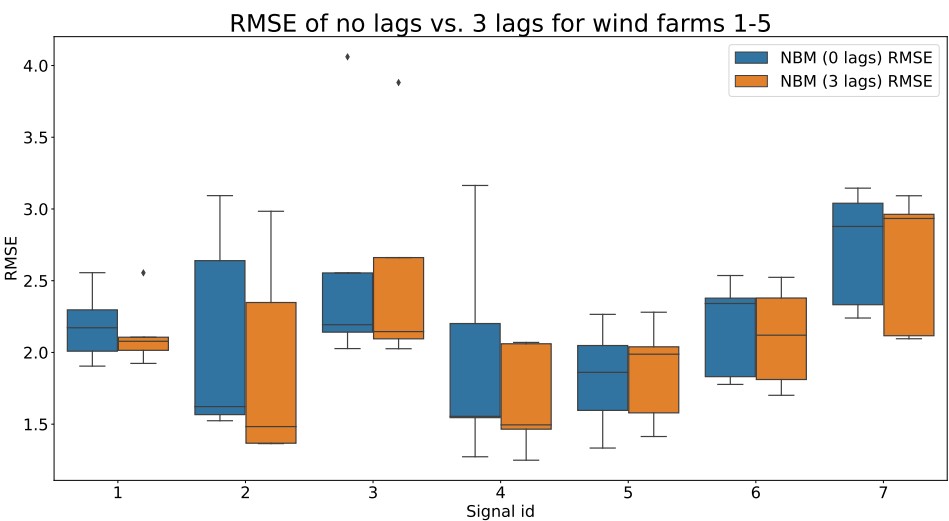

**Figure 11.** Comparison of RMSE of the no lags and the 3 lags NBM for wind farms 1-5. Signal ID 1 = TempGearbBear_1 (avg), signal ID 2 = TempGearbBear_2 (avg), signal ID 3 = TempGearbInlet (avg), signal ID 4 = TempGenBearing_1 (avg), signal ID 5 = TempGenBearing_2 (avg), signal ID 6 = TempRotorBearing (avg), signal ID 7 = TempStatorWind (avg).

at the first generator bearing. For the generator bearing failure the $\Delta\text{PE}_{unh-h}$ is clearly positive for $T_{gen\_bear\_2}$. This is also in line with the expectations. Since $\Delta\text{PE}_{unh-h}$ is only positive for the second generator bearing and not for the first generator bearing, it is likely that most of the bearing failures happened at the second generator bearing. Unfortunately, the replacement information received from the wind turbine operator does not indicate which bearing has failed, so this statement can not be

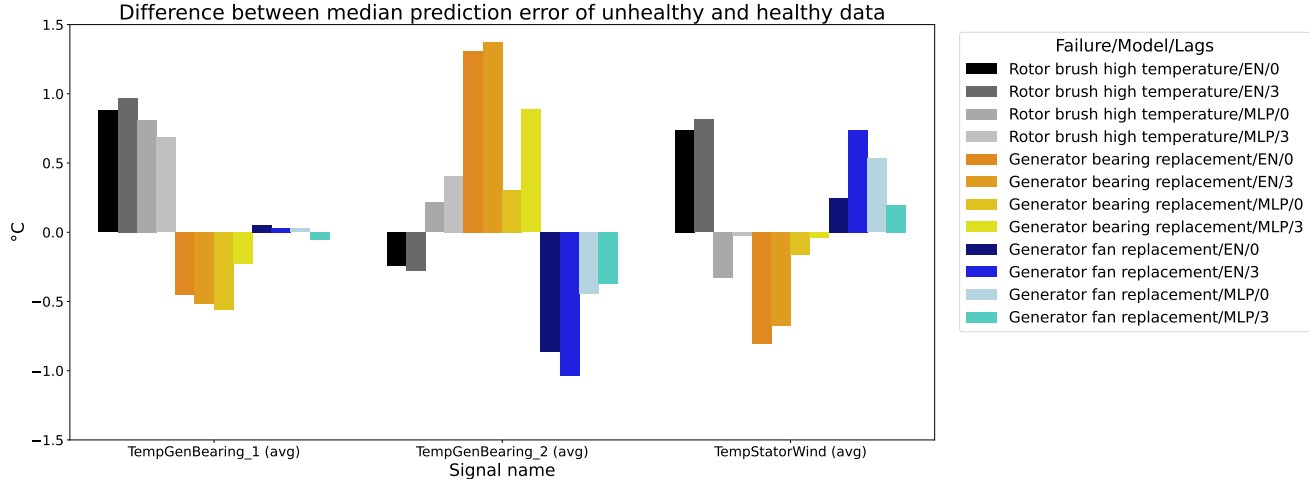

**Figure 12.** Difference between the median prediction error for the unhealthy and healthy data for elastic net and MLP models with no or 3 lags of the predictor variables. EN = elastic net, MLP = multi-layer perceptron, 0 = no predictor lags, 3 = 3 predictor lags ($x_{t-1}, x_{t-2}, x_{t-3}$).

verified. For the generator fan failures the $\Delta \text{PE}_{unh-h}$ is clearly positive for the elastic net with three lags. This is not the case for the other signals, which is quite surprising since the hypothesis is that a generator fan failure should be indirectly visible in all three generator signals. Furthermore, there is no evidence that the MLP (with 0 or 3 lags) improves upon the elastic net model. The meaning of this will be discussed below. The results in Figure 12 give only a first indication of whether the NBM is useful for anomaly detection or not. The lack of a clear positive $\Delta \text{PE}_{unh-h}$ for signals where it is expected to be positive does not mean that all is lost. The anomaly detection techniques discussed in Experiment 6 are more sensitive for small deviations. This makes that they can in some cases still detect anomalies even though the $\Delta \text{PE}_{unh-h}$ is not clearly positive.

From the results of the second experiment, it can be concluded that the addition of 3 lags, only marginally improves the model accuracy on healthy data. There is some evidence that the addition of the predictor lags results in NBMs with more anomaly detection potential. However, the improvement is in general small. Taking into account that adding the lags of all the predictors results in a strong increase in the dimensionality of the problem and the computational time, it is debatable whether the (limited) performance gains outweigh the extra cost. Reasons for the low added value of the lags can perhaps be an insufficient number of lags, a lack of information on the dynamics in the aggregated SCADA data, or the combination of transient and non-transient behavior. The first hypothesis seems to be unlikely since limited experimentation using more lags showed no clear improvement in performance. The second hypothesis is possible due to the fact that subtracting the fleet median from the SCADA data signals (see preprocessing section) results in less autocorrelation in the data. Also, the aggregation of the data to hour level will have an impact. This results in lagged predictor values that are less informative. The third hypothesis would imply that the dynamics of the steady-state and the transient behavior of the turbine are so different they can not be learned by one elastic net model. This is a possible explanation. A solution to this problem would be to train a separate model for the transient and steady-state behavior or use a more complex model that is better able to learn the differences between the

two states. The fact that no performance gains are achieved when using the MLP makes this hypothesis however somewhat less convincing. However, to give a conclusive answer to which hypothesis is the correct one, further research is required.

## 4.3 Experiment 3: Impact of reducing the training data to 2 months per turbine instead of 6

Pipeline configuration: implicit NBM based on fleet median, explicit NBM based on elastic net regression, heuristic-based healthy data selection, full grid search hyperparameter tuning (5-fold CV), 2 months training data per turbine.

In this experiment, the impact is analyzed of reducing the amount of training data from 6 months per turbine to 2 months. This is relevant since less training data means less computational and startup time when using the pipeline on a new wind farm. However, reducing the amount of training data in general also comes at a cost. The model accuracy tends to decrease. The question is how much and whether it weighs up against the advantages. Furthermore, since the training data is selected in chronological order (meaning the first X healthy observations), less training data means that it becomes more likely that certain turbine conditions are missed (or are underrepresented in the data). This can for example be the case with long-duration power downs which cause exceptionally low temperatures for certain components. The most likely result will be that those conditions will be less well modeled, resulting in a larger prediction error. Depending on the use case of the pipeline this might be a problem.

Figures 13 and 14 show indeed that there is some loss of prediction accuracy when the amount of training data is reduced from 6 to 2 months. This shows itself as an increase in the prediction error. During steady-state behavior, this loss is not really visible, but during transient behavior, the loss of fit can be substantial (see for example Figure 13). This is most likely caused by the fact that the training data does not (sufficiently) contain similar transient behavior examples. Figure 15 gives a more general overview of the RMSE results. It shows that the reduction in training data in general leads to an increase in the healthy test data RMSE. This increase is not massive, but often also not negligible. For some signals, the median RMSE is slightly smaller. This reduction should not be considered evidence for a superior model but more as an indication that the influence of the sample on the results is considerable. This is something that should be taken into account when analyzing the results.

Figure 16 shows the differences between the median prediction errors on the unhealthy and healthy data ($\Delta\text{PE}_{unh-h}$). The results indicate that for detecting rotor brush high temperature failures, 6 months of training data is better than 2. This is clear from the fact that $\Delta\text{PE}_{unh-h}$ for $T_{stator}$ is much larger when using 6 months of training data. The generator bearing failures are detected clearly in $T_{gen\_bear\_2}$. Again it can be observed that $\Delta\text{PE}_{unh-h}$ is somewhat smaller when using only 2 months of training data. The analysis for the generator fan failures is somewhat less clear. On the one hand, there is, surprisingly, a larger $\Delta\text{PE}_{unh-h}$ for the model trained on only two months of data when using $T_{gen\_bear\_2}$ as the target signal. On the other hand, the model trained on 2 months of training data does not result in a positive $\Delta\text{PE}_{unh-h}$ for the $T_{stator}$ signal, contrary to the model with 6 months of training data. Overall it can be stated that less training data results in general in NBMs with less potential to be useful for anomaly detection.

Experiment 3 has shown that reducing the training data to 2 months results in reduced performance of the model. This is not so much a problem for the behavior states that are frequently shown by the turbine (e.g. steady-state behavior). It does have however a large influence on the states that are rare (e.g. long-term cool-downs, ...). Furthermore, the results also show

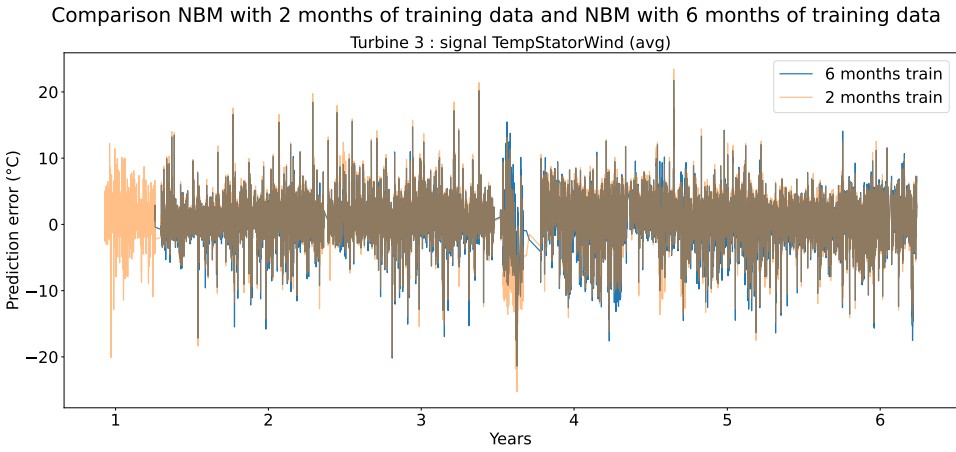

**Figure 13.** Prediction error NBMs with 6 months and 2 months of training data for the TempGenBearing_2 (avg) signal of turbine 3 of wind farm 5

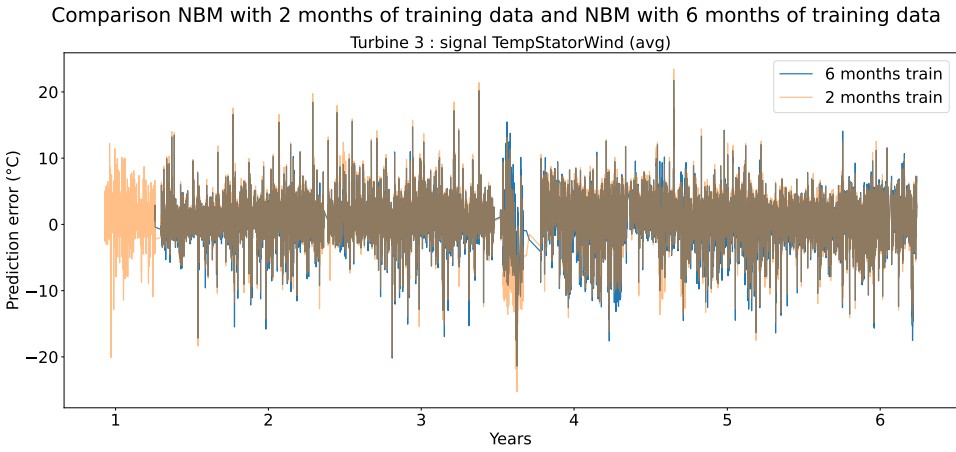

**Figure 14.** Prediction error NBMs with 6 months and 2 months of training data for the TempStatorWind (avg) signal of turbine 3 of wind farm 5

that the anomaly detection potential of the NBM decreases if the amount of training data is reduced to 2 months. Whether the reduced performance of the NBM is a problem, really depends on the use case. For some use cases, the larger prediction error in certain rare states is not an issue. For those cases, it might be useful to reduce the amount of training data because it will reduce the computational burden of the pipeline. However, if rare behavior is important, the advantages may not outweigh the disadvantages.

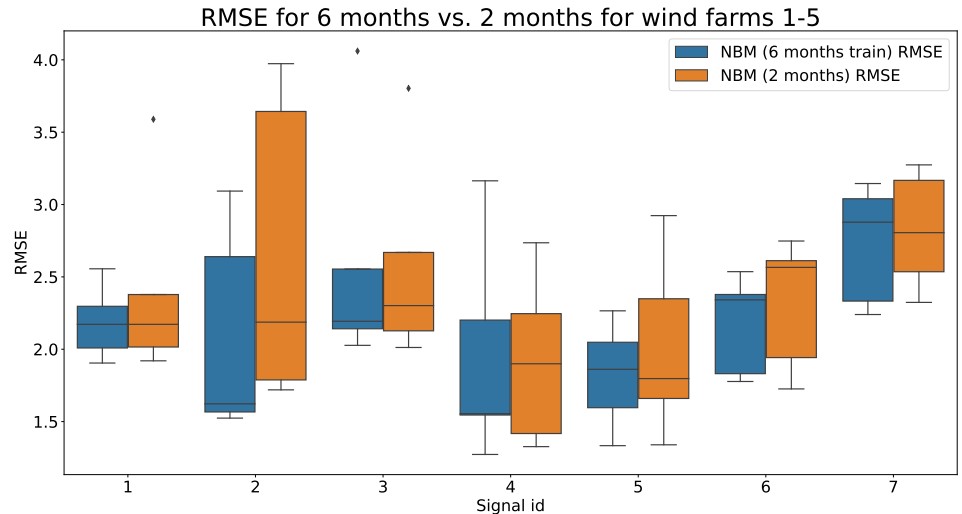

**Figure 15.** Comparison of RMSE when using 6 months or 2 months of training data for the NBM model for wind farms 1-5. Signal ID 1 = TempGearbBear_1 (avg), signal ID 2 = TempGearbBear_2 (avg), signal ID 3 = TempGearbInlet (avg), signal ID 4 = TempGenBearing_1 (avg), signal ID 5 = TempGenBearing_2 (avg), signal ID 6 = TempRotorBearing (avg), signal ID 7 = TempStatorWind (avg).

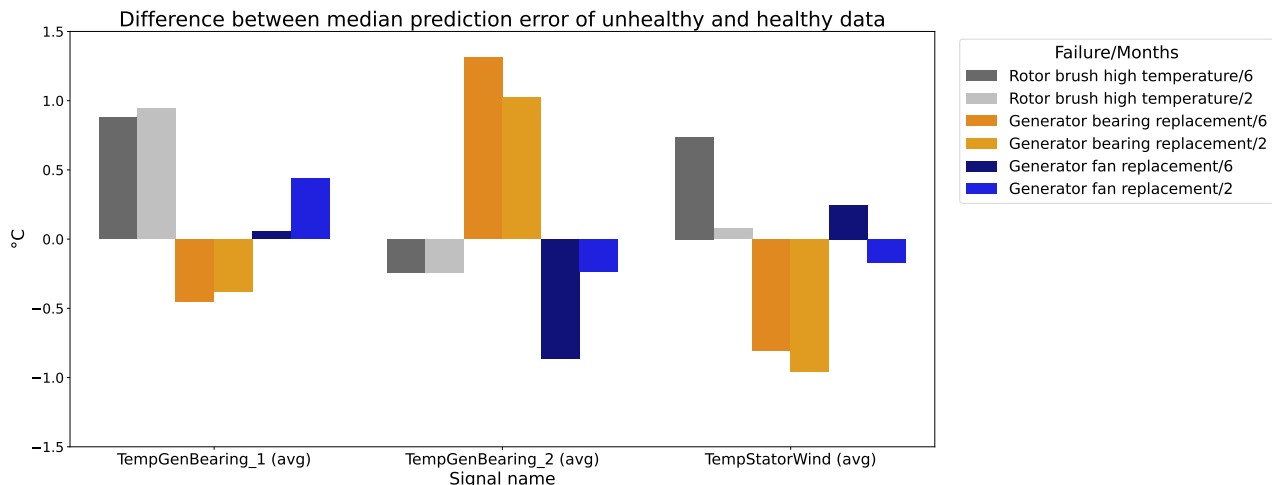

**Figure 16.** Difference between the median prediction error for the unhealthy and healthy data ( $\Delta\mathrm{PE}_{unh-h}$ ) for elastic net models with 6 months and 2 months of training data. 6 = 6 months of training data, 2 = 2 months of training data. The results are shown for TempGenBearing_1 (avg) ($T_{gen\_bear\_1}$), TempGenBearing_2 (avg) ($T_{gen\_bear\_2}$) and TempStatorWind (avg) ($T_{stator}$).

## 4.4 Experiment 4: Added value of a PCA transformation step before the explicit NBM modeling

Pipeline configuration: implicit NBM based on the fleet median, PCA transformation, explicit NBM based on elastic net regression, heuristic-based healthy data selection, full grid search hyperparameter tuning (5-fold CV), 6 months of training data per turbine.

In the fourth experiment, the impact of PCA transforming (only a transformation, no dimensionality reduction) the data prior to the elastic net modeling is analyzed. Normally the elastic net algorithm should be able to handle high dimensional data with strong correlations between some of the predictors. However, in practice, there might still be some benefit of first PCA-transforming the data. Figure 19 shows that the prediction accuracy in general does not (or only marginally) improve when a PCA transformation step is added to the pipeline. For several signals, the opposite happens. However, these results hide certain interesting side effects. Figure 17 shows that the pipeline with the PCA tends to be a better modeler of the cool-downs than the model without the PCA. Figure 18 also shows that the pipeline without the PCA in some rare cases generates (unrealistically) large prediction errors, while this is not the case when the PCA is used. Furthermore, the training time (and hyperparameter tuning) is considerably shorter for the pipeline with the PCA, even though the number of combinations being tested during tuning is much larger (2728 for the pipeline with PCA, 341 for the pipeline without PCA). This might have something to do with the fact that the new features generated by the PCA transformation are uncorrelated. This can improve the training of the elastic net. For example, for wind farm 5 the hyperparameter tuning without PCA took 4329 s, while with the PCA it took only 3859 s.

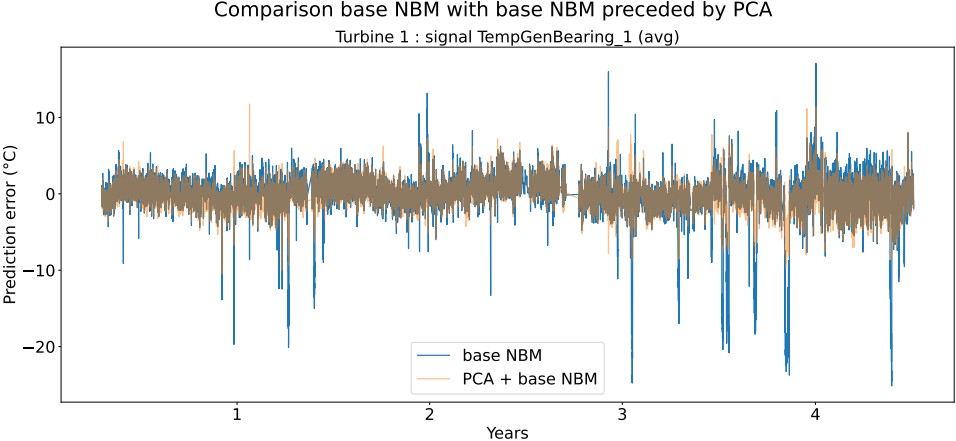

**Figure 17.** Prediction error base NBM and PCA + NBM for the TempGenBearing_1 (avg) signal of turbine 1 of wind farm 1

Figure 20 shows the differences between the unhealthy and the healthy prediction error ($\Delta\mathrm{PE}_{unh-h}$). The results show that using the PCA as a preprocessing step has in general a small positive impact on $\Delta\mathrm{PE}_{unh-h}$. For the rotor brush high temperature failures there is no clear change in the positive difference for $T_{stator}$ signal for the model with and without PCA

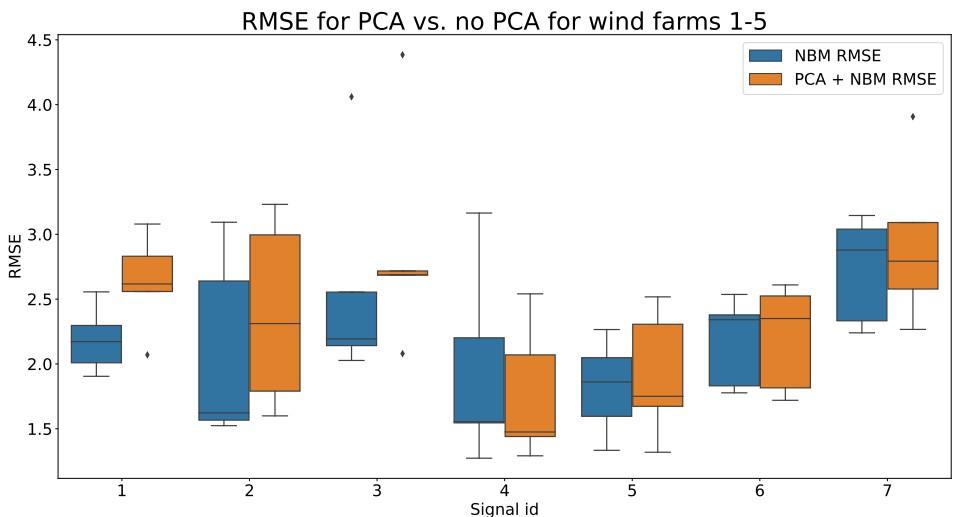

**Figure 18.** Prediction error base NBM and PCA + NBM for the TempGenBearing_2 (avg) signal of turbine 3 of wind farm 3

**Figure 19.** Comparison of the RMSE when the data is PCA transformed to when the data is not PCA transformed for wind farms 1-5. Signal ID 1 = TempGearbBear_1 (avg), signal ID 2 = TempGearbBear_2 (avg), signal ID 3 = TempGearbInlet (avg), signal ID 4 = TempGenBearing_1 (avg), signal ID 5 = TempGenBearing_2 (avg), signal ID 6 = TempRotorBearing (avg), signal ID 7 = TempStatorWind (avg).

preprocessing. For the generator bearing failures the positive difference for the $T_{gen\_bear\_2}$ signal is slightly larger when using the model with PCA preprocessing. The same is true for the $T_{stator}$ signal for the generator fan failures.

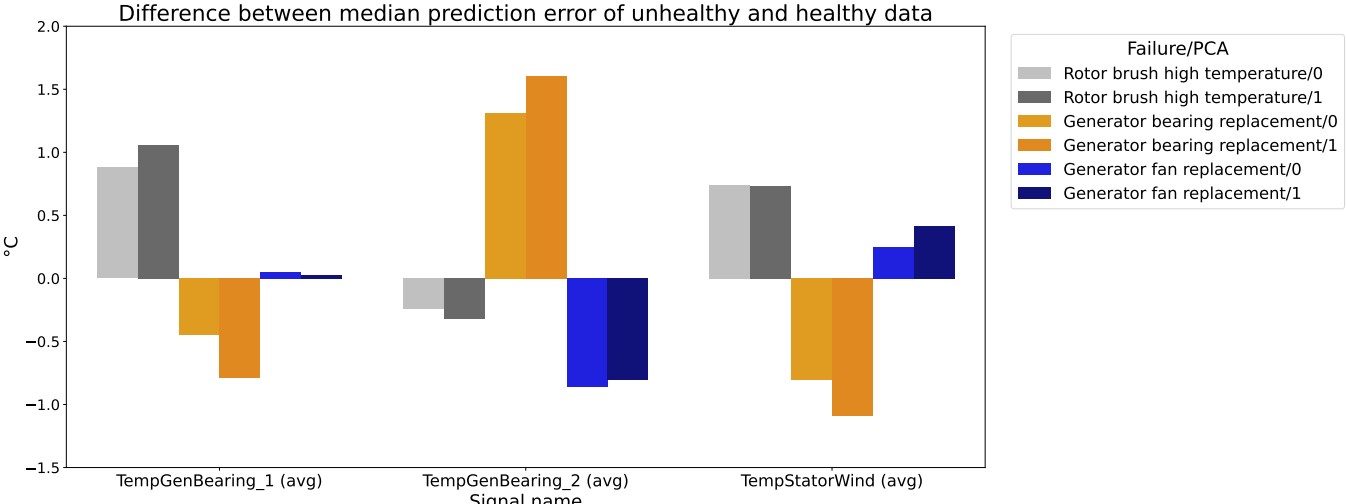

**Figure 20.** Difference between the median prediction error for the unhealthy and healthy data for elastic net models with and without PCA preprocessing. 0 = no PCA preprocessing, 1 = PCA preprocessing. The results are shown for TempGenBearing_1 (avg) ($T_{gen\_bear\_1}$), TempGenBearing_2 (avg) ($T_{gen\_bear\_2}$) and TempStatorWind (avg) ($T_{stator}$).

Based on the results of the fourth experiment, it can be concluded that PCA transforming the data prior to the elastic net modeling can be beneficial in some cases. Although the healthy test data RMSE does not decrease, it does avoid certain large (erroneous) prediction errors and it seems to improve the modeling of the transient behavior. It also seems to improve somewhat the distinction the NBM can make between healthy and unhealthy data. Furthermore, it also reduces the training time considerably. This means that PCA transforming the data prior to the elastic net modeling has some benefits. Whether these weigh up against the added complexity and the loss of the original features (the principal components are linear combinations of all the original features) depends of course on the case.

### 4.5 Experiment 5: Added value of more complex NBM models

Pipeline configuration: implicit NBM based on the fleet median, explicit NBM based on elastic net regression or SVR or light GBM or MLP, heuristic-based healthy data selection, full grid search hyperparameter tuning for elastic-net, SVR and light GBM and randomized grid search tuning for MLP (5-fold CV), 6 months of training data per turbine.

The fifth experiment focuses on the complexity of the relations between the inputs (predictors) and the outputs (targets) of the NBM problem. The elastic net regression is a relatively simple model that works well when the problem is linear (meaning linear in the parameters). However, if the problem is highly non-linear this type of model is not really suitable. In the latter case, more complex models like tree-based algorithms or neural networks are more appropriate. The trade-off is however that these models are more "black box" and require in general much more training data. Nevertheless, it is still interesting to analyze the performance of these models. If they clearly outperform the elastic net model then that is evidence for the existence of

non-linear relations. The performance of the elastic net will be compared with that of an SVR with a radial kernel (also used in (Castellani et al., 2021)), a light GBM (which is similar to the Gradient Boosting algorithms used in (Udo and Yar, 2021), (Maron et al., 2022) and (Beretta et al., 2021)) and a MLP.

Figure 21 shows that the prediction error for the four models is roughly similar in size, with some large upward or downward spikes at certain points in time for the SVR. Figure 22 shows however that under certain conditions the SVR and light GBM performance severely degrades. This is the case during large long-term cool-downs. This problem also impacts the elastic net and the MLP, but to a lesser extent. With limited examples in the training dataset, the SVR and light GBM have difficulties estimating the normal behavior in those cases. The elastic net, which is a much simpler model, might use relatively accurate extrapolations for estimating those cases. The MLP performs during cool-downs clearly better than the SVR and light GBM, but does not completely reach the performance of the elastic net. A possible explanation is that the MLP is better at learning the non-linearities than the SVR and light GBM, but the problems it has with extrapolation, make that it does not perform as well as the elastic net. Figure 23 gives an overview of the healthy test data RMSEs. The results show that the SVR never performs best, and it often performs significantly worse than the three other models. The light GBM and MLP perform for some signals marginally better than the elastic net, however, the improvement is small. For some signals, they perform worse than the elastic net. This means that there is some evidence for non-linearities when modeling some of the signals. The evidence is however weak.

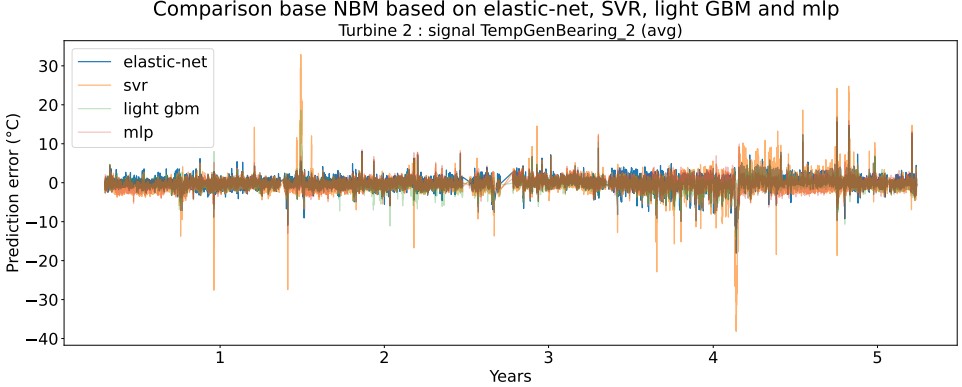

**Figure 21.** Prediction error elastic net, SVR, light GBM and MLP NBMs for TempGenBearing_2 (avg) signal of turbine 2 of wind farm 5

Figure 24 shows the differences between the unhealthy and the healthy prediction error ($\Delta\text{PE}_{unh-h}$). The purpose of this plot is to see whether using more complex machine learning models results in NBMs that are better at distinguishing healthy from unhealthy data. For the rotor brush high temperature failures the elastic net model outperforms the other models. This is clear from the fact that the $\Delta\text{PE}_{unh-h}$ for the $T_{stator}$ signal is the largest for the elastic net. For the generator bearing failure the story is the same. The elastic net outperforms the other models. The prediction error difference for the $T_{gen_bear_2}$ signal is the largest for the elastic net. For the generator fan failures the results indicate that the more complex models somewhat outperform the elastic net model. The light GBM has a larger $\Delta\text{PE}_{unh-h}$ for the $T_{gen_bear_1}$ signal, and the MLP and SVR

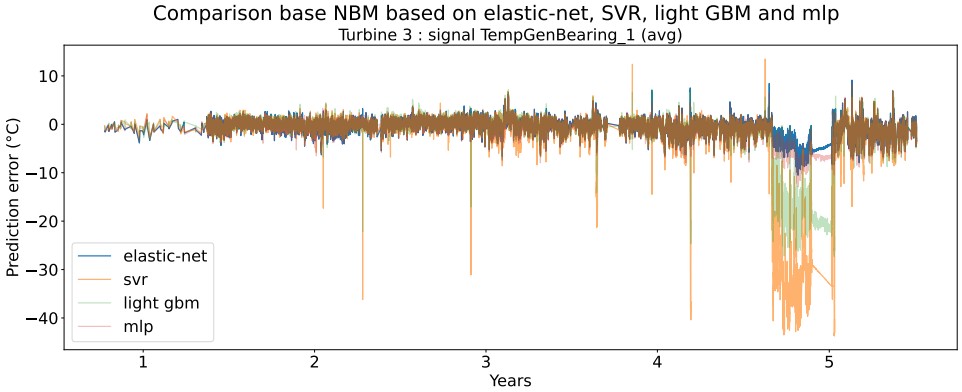

**Figure 22.** Prediction error elastic net, SVR, light GBM and MLP NBMs for TempGenBearing_1 (avg) signal of turbine 3 of wind farm 2

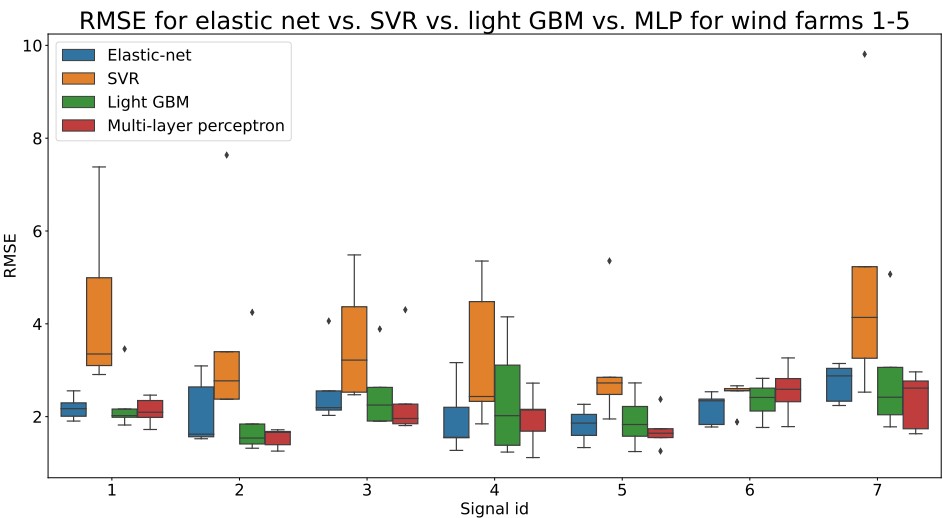

**Figure 23.** Difference between the median prediction error of the unhealthy and healthy data. Comparison of the RMSE of the elastic net, the SVR, the light GBM, and the MLP NBM models for wind farms 1-5. Signal ID 1 = TempGearbBear_1 (avg), signal ID 2 = TempGearbBear_2 (avg), signal ID 3 = TempGearbInlet (avg), signal ID 4 = TempGenBearing_1 (avg), signal ID 5 = TempGenBearing_2 (avg), signal ID 6 = TempRotorBearing (avg), signal ID 7 = TempStatorWind (avg).

have a larger $\Delta\mathrm{PE}_{unh-h}$ for the $T_{stator}$ signal. These results indicate that using a combination of several of the more complex models might result in a better anomaly detection performance.

The results of experiment 5 show some (weak) evidence that the relation between the predictors and some of the targets is non-linear. This means that some non-linear models (e.g. from the traditional machine learning or the deep learning domain) might improve the modeling of the healthy data. The improvements obtained by the light GBM and the MLP are however small.

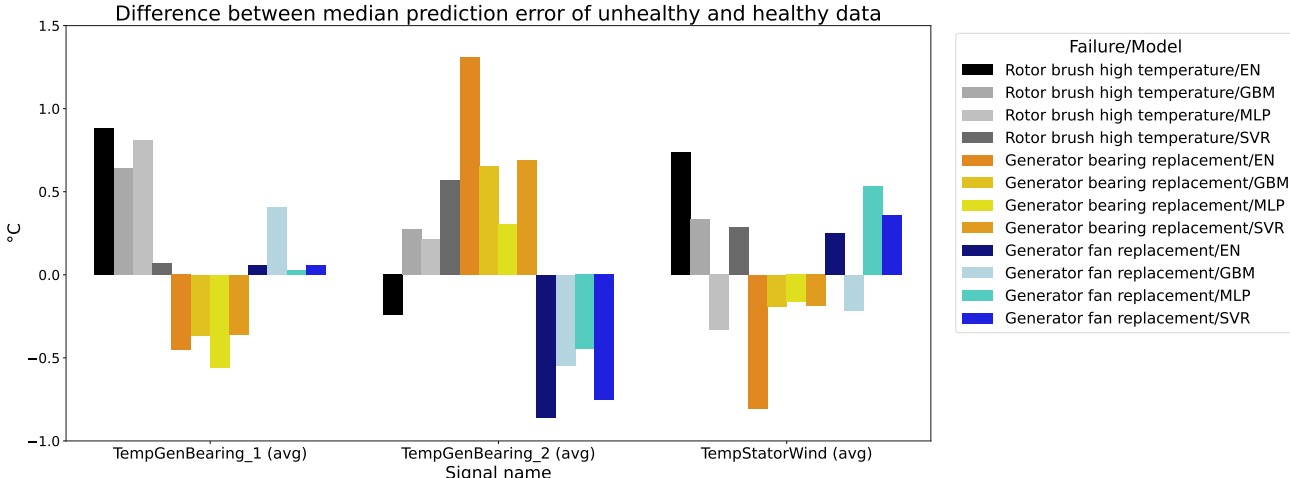

**Figure 24.** Difference between the median prediction error for the unhealthy and healthy data for elastic net, light GBM, multi-layer percep­tron and support vector regression models. EN = elastic net, GBM = light-gradient boosting machine, MLP = multi-layer perceptron, SVR = support vector regression. The results are shown for TempGenBearing_1 (avg) ($T_{gen\_bear\_1}$), TempGenBearing_2 (avg) ($T_{gen\_bear\_2}$) and TempStatorWind (avg) ($T_{stator}$).

Furthermore, if the $\Delta PE_{unh-h}$ is analyzed then there is no evidence that the more complex models outperform the elastic net for rotor brush high temperature and generator bearing failures. However, for generator fan failures it might be beneficial to use the more complex models or a combination of the models. Furthermore, the results also show that the more complex models are more susceptible to the underrepresentation of certain states in the training dataset. This can lead to severe performance degradation.

### 4.6 Experiment 6: Identifying anomalies in the prediction error using Iterative Outlier Detection and CUSUM

The sixth and last experiment focuses on detecting anomalies in the prediction error of the NBM. The previous experiments all focused on the NBM itself because it is the basis of the anomaly detection pipeline. In this section, the focus shifts to the anomaly detection algorithms that can be used to find abnormal prediction error patterns. As shown in the state-of-the-art section there are multiple ways how this can be achieved. Testing and comparing all the methods that have been developed is obviously unfeasible. For this reason, a selection will be made that takes into account the requirements of the industry, namely maintainability, transparency, computational efficiency, and robustness. Preference is given to univariate statistical techniques that have been thoroughly studied, i.e. the IOD-MAD and the CUSUM. The accuracy of the techniques will be, as described in the methodology section, assessed both in a quantitative and qualitative fashion. For practical reasons, the figures shown in this section are only a subset of all figures that can be generated from the results.

### 4.6.1 Generator bearing replacement

Generator bearing failures can normally be detected in the temperatures of the bearings. When damage or wear is forming the temperatures start to increase due to increased friction. The information from the wind turbine operator, unfortunately,
does not mention which bearing was replaced, e.g. bearing 1 or 2. However, in general, this can be deduced from the results because the health degradation is much more pronounced for one of the two bearings. Generator bearing failures normally form slowly over time. This means that health degradation shows itself mostly during a prolonged period of time. The strength of the degradation depends however also on how fast the bearing was replaced. If it was replaced as part of preventive maintenance then the degradation signal will most likely be less strong compared to situations in which the bearing truly failed. In total
information on 4 generator bearing replacements is available.

The generator bearing replacement in Figures 25 and 26 is detected by the IOD-MAD and the CUSUM algorithms, although the latter detects it only lightly. The IOD-MAD tends to generate more anomalies that can not be associated with this failure than the CUSUM. The generator bearing replacement in Figures 27 and 28 is also clearly detected by both the IOD-MAD and the CUSUM algorithm. The fact that both algorithms detect the replacement strongly indicates most likely that the degradation
of the bearing was severe. Both algorithms raise around year 5 also some anomalies. It is unclear at the moment what the reason for this is.

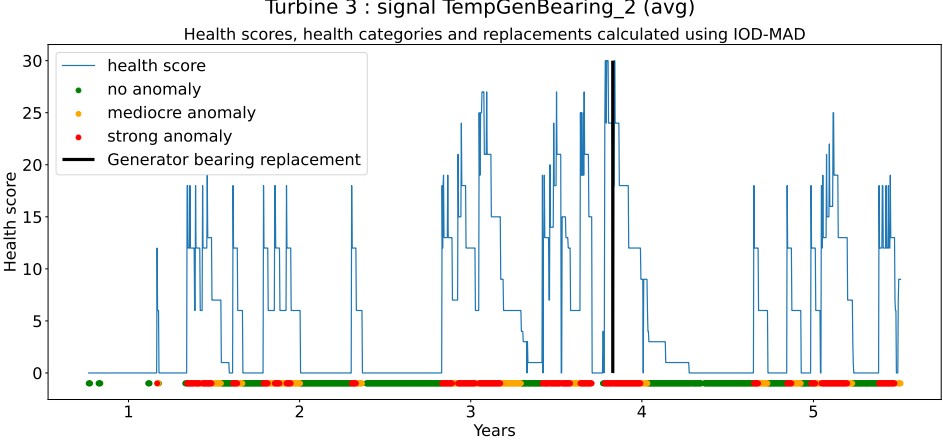

**Figure 25.** Health score and category based on IOD-MAD algorithm for TempGenBearing_1 (avg) of turbine 3 of wind farm 3.

The overall results show that the IOD-MAD algorithm is able to identify 3 out of 4 generator bearing replacements, while the CUSUM found only 2 out of 4. The IOD-MAD has however the tendency to generate more bad health flags that can not be associated with the failures. This probably means that the false positive rate for this algorithm is higher than for the CUSUM.
The CUSUM on the other hand seems to be much less sensitive (probably not sensitive enough). It is likely that changing the hyperparameters would improve the performance. However, parameter tuning with such a small number of examples is difficult.

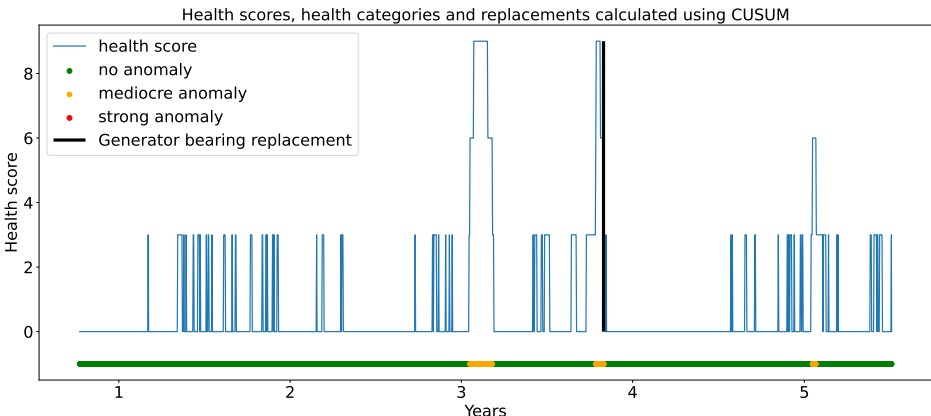

**Figure 26.** Health score and category based on CUSUM algorithm for TempGenBearing_1 (avg) of turbine 3 of wind farm 3.

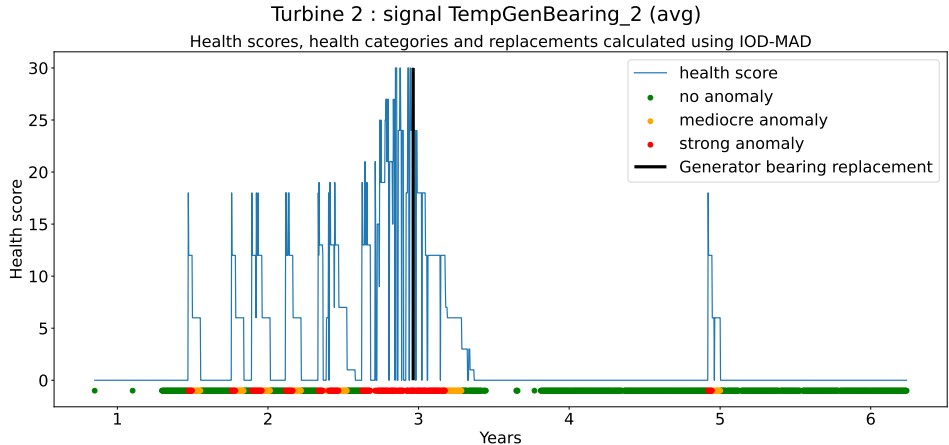

**Figure 27.** Health score and category based on IOD-MAD algorithm for TempGenBearing_1 (avg) of turbine 2 of wind farm 4.

### 4.6.2 Generator fan replacement

Generator fan problems can only be observed indirectly for the turbines of wind farms 1-5. This is due to the fact that there
is no signal available in the SCADA data that is directly linked with the fans. However, it can be assumed that the failure of
a generator can be observed indirectly by analyzing the temperatures of generator components. Even though this should be
possible, indirect observations are probably less clear than direct observations, meaning that the health degradation will most
likely be much less clear. Initially, it was assumed that a generator fan failure could be identified if all three generator signals,
e.g. TempGenBearing_1 (avg), TempGenBearing_2 (avg), and TempStatorWind (avg), show health degradation. However,
in practice, it appears that this is not the case and that in general the failure can only be spotted in one signal, namely the

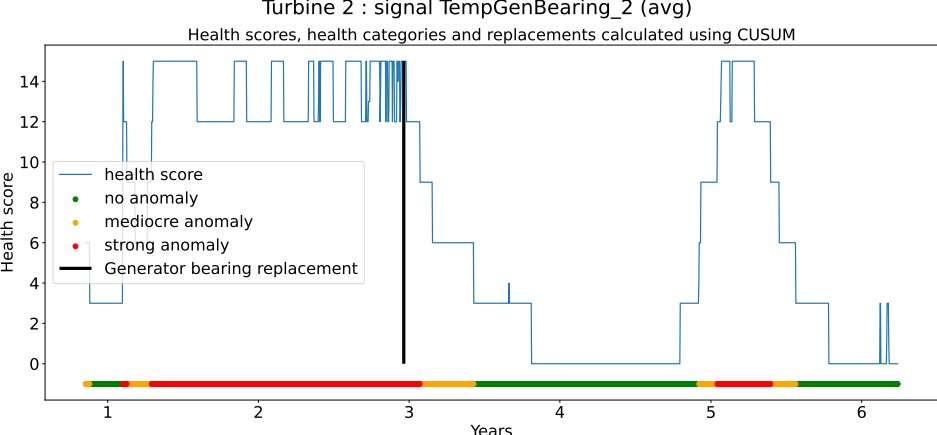

**Figure 28.** Health score and category based on CUSUM algorithm for TempGenBearing_1 (avg) of turbine 2 of wind farm 4.

TempGenBearing_1 (avg) signal. The result of this is that it is often difficult to make a distinction between a generator bearing failure and a generator fan failure. This means that when bad health is observed in the TempGenBearing_1 (avg) signal, the user will receive the warning that there is an issue with the bearing or the fan.

The datasets for wind farms 1-5 contain in total 3 usable examples. The three examples show that the IOD-MAD finds all the replacements correctly. The CUSUM misses one. Figures 29 and 30 show the results for the third generator fan failure in the dataset. The IOD-MAD algorithm also raises bad health flags at some other points in time. This is less the case for the CUSUM. It would be a bit premature to immediately decide that those are false positives. After all the TempGenBearing_1 (avg) signal can also be influenced by issues with the bearings, or some other factors. In practice, it would mean that at those points in time the user would also get a warning for a potential fan or bearing failure. However, the number of those cases is relatively small given that the observation window is nearly 6 years long.

Overall the IOD-MAD algorithm identified three out of three generator fan replacements correctly, while the CUSUM found two out of three. The results show also that the CUSUM is more conservative than the IOD-MAD, which makes it less capable. However, it also generates fewer bad health flags that can not be linked to the generator fan replacements.

### 4.6.3   Rotor brush high temperature failure

The datasets for wind farms 1-5 contain 5 examples of rotor brush high temperature failures that can be used for the validation of the anomaly detection models. 2 failures had to be excluded due to missing SCADA data. Just like the generator fan replacements, there is no direct way to identify this failure. It is however assumed that they can be identified indirectly through the temperature of the stator windings (although there is some debate whether this is always the case). Just like for the generator fan replacement case, it means that the signal might not always be very strong.

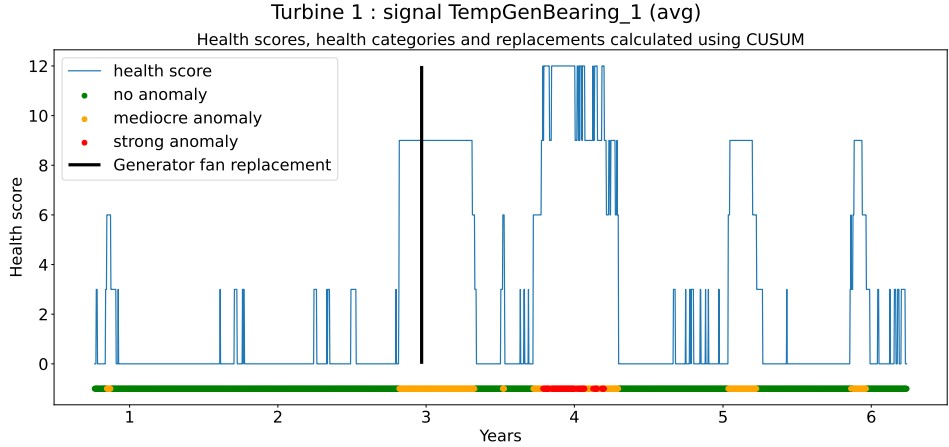

**Figure 29.** Health score and category based on IOD-MAD algorithm for TempGenBearing_1 (avg) of turbine 1 of wind farm 5.

**Figure 30.** Health score and category based on CUSUM algorithm for TempGenBearing_1 (avg) of turbine 1 of wind farm 5.

The first rotor brush high temperature failure (Figures 31 and 32) is detected by both algorithms. For the IOD-MAD the main detections happen after the date of the replacement, but given the uncertainty about the event dates, and the fact that sometimes an issue is not solved after the first try, it can still be considered a correct detection. The detection by the CUSUM algorithm is very clear. The generator rotor brush high temperature failure in Figures 33 and 34 are also detected correctly. However, the detection strength is lower for the CUSUM algorithm than for the IOD-MAD algorithm.

The overall results show that both algorithms can quite reliably detect the rotor brush high temperature failures using the temperature of the stator windings. Four out of five failures were detected. However, from the results, it is also clear that the detections are not always very clear. Nevertheless, they are still sufficiently different from the rest of the data to be useful for the user.

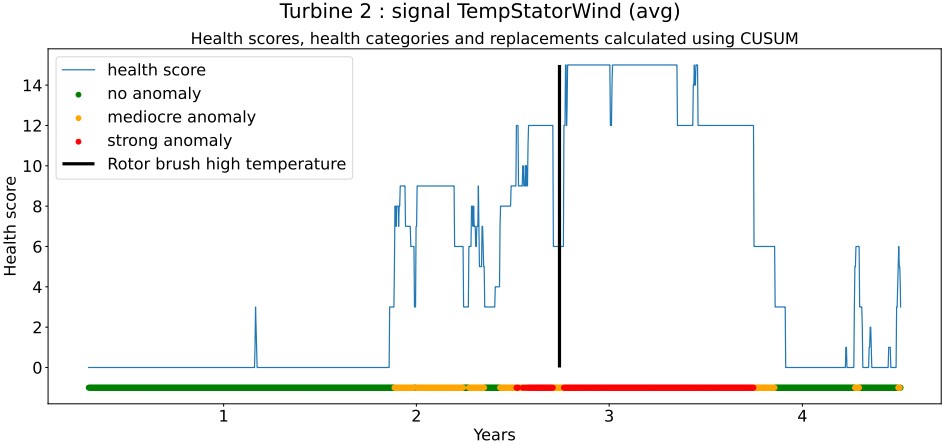

**Figure 31.** Health score and category based on IOD-MAD algorithm for TempStatorWind (avg) of turbine 2 of wind farm 1.

**Figure 32.** Health score and category based on CUSUM algorithm for TempStatorWind (avg) of turbine 2 of wind farm 1.

### 4.6.4 False positive ratio

The last part of this experiment is assessing the false positive ratio of the anomaly detection models. The analysis will be focused on the IOD-MAD algorithm because it has the highest detection accuracy. The results in Table 2 show that false positive ratios for the different signals are, 0.12 for the TempGenBearing_1 (avg) signal and 0.08 for the TempGenBearing_2 (avg) and the TempStatorWind (avg) signals. This is a relatively small ratio given that there is always some uncertainty about the health of the data. The results show however that for some turbines the false positive ratio for certain signals can be high. This is for example the case for the TempGenBearing_2 (avg) signal of turbine 1 of wind farm 2. This might indicate that there

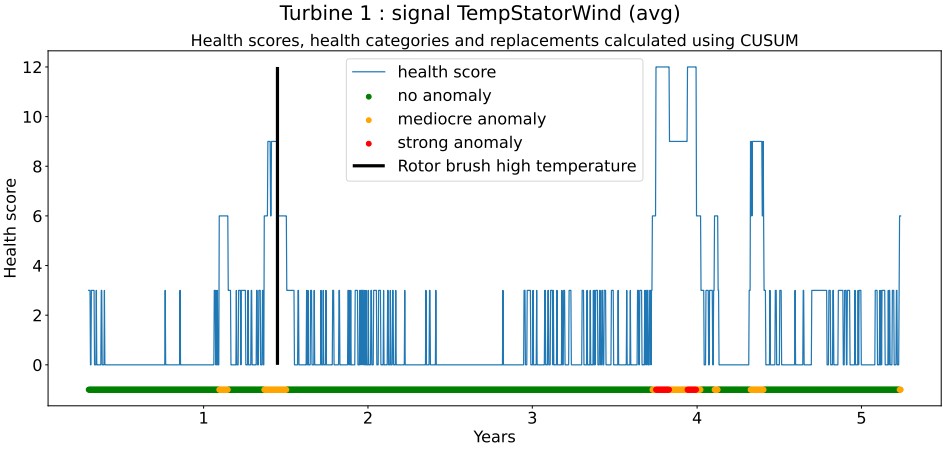

**Figure 33.** Health score and category based on IOD-MAD algorithm for TempStatorWind (avg) of turbine 1 of wind farm 4.

**Figure 34.** Health score and category based on CUSUM algorithm for TempStatorWind (avg) of turbine 1 of wind farm 4.

is a hidden underlying issue with the second bearing of this turbine. Nevertheless, it can be concluded that in general, the false positive ratio is quite low.

## 5    Conclusion and future research

This paper gives an overview of recent research on condition monitoring of wind turbines using SCADA data and the NBM framework. The goal is to give the reader an idea of what the current state-of-the-art is, e.g. what has been tried, and what the performance of techniques is on data from operational wind farms. This is done by first presenting a structured overview of the current state-of-the-art. This gives an idea of how an NBM pipeline is normally set up and which techniques are used for

| WF / T | TempGenBearing_1 (avg) | TempGenBearing_2 (avg) | TempStatorWind (avg) |
|--------|------------------------|------------------------|----------------------|
| 1 / 1 | 0.03 | 0.03 | 0.14 |
| 1 / 4 | 0.09 | 0.09 | 0.06 |
| 2 / 1 | 0.28 | 0.43 | 0.03 |
| 3 / 1 | 0.12 | 0.25 | 0.08 |
| 3 / 2 | 0.15 | 0.04 | 0.08 |
| 3 / 4 | 0.02 | 0.08 | 0.02 |
| 4 / 3 | 0.14 | 0.17 | 0.01 |
| 4 / 4 | 0.12 | 0.24 | 0.11 |
| 5 / 2 | 0.16 | 0.04 | 0.11 |
| 5 / 3 | 0.06 | 0.05 | 0.09 |
| **Median** | **0.12** | **0.08** | **0.08** |

**Table 2.** Overview of the ratio of bad health observations on the total number of observations. WF stands for wind farm and T for turbine.

the different steps. In the second part of the paper, several techniques from the state-of-the-art are selected and applied to data from several real operational wind farms. This is done through six demonstration experiments. The different techniques are compared, and their performance is thoroughly analyzed. Five experiments focus on the NBM model, one experiment focuses on the analysis of the prediction error.

The first experiment examines the modeling performance of a relatively simple NBM model, i.e. the elastic net. The results show that the model is a capable modeler, even during the transient behavior of the turbine. The second experiment discusses the impact of using the lagged values of the predictors as input to the elastic net model. The results show only a marginal improvement of the model quality (minor reduction in the healthy test data RMSE). The modeling of the transient behavior is not noticeably better. Potential explanations for this are, firstly, data with a 1-hour resolution might be insufficient, secondly, 3 lagged values might not be enough, and thirdly, a single model for transient and non-transient behavior might be problematic due to differences in the behavior dynamics. The third experiment examines the impact of reducing the amount of training data from 6 to 2 months. The results show a reduction in the modeling accuracy (as expected). The risk of underrepresentation of certain wind turbine states in the training data also increases. This can result in degraded model performance for these states. The fourth experiment discusses the added value of PCA transforming the output of the implicit NBM before it is given to the elastic net. The results show that the overall model performance does not improve when the PCA transformation is used. However, some abnormal large prediction errors disappear and the run time of the pipeline is significantly reduced. A possible explanation for this is the fact that the PCA breaks the correlation between the predictors, which results in a more stable model and faster convergence. The fifth experiment investigates whether more complex machine learning models are useful for the pipeline. The results show that the SVR with a radial kernel performs overall worse than the elastic net, while the light GBM and MLP performs slightly better (for some signals). However, these more complex models can suffer from severe model degradation during the transient behavior of the turbine. A possible explanation can be found in the underrepresentation of this

behavior in the training dataset. A possible solution for this might be oversampling the minority behavior. The sixth and last experiment tests two different univariate anomaly detection techniques (IOD-MAD and CUSUM) that generate a health score for the signals. The results show that IOD-MAD is able to identify the failures more accurately, at the cost of more alerts during periods that not immediately can be linked to a failure. Most generator bearing replacements, generator fan replacements and rotor brush high temperature failures can be detected accurately. Furthermore, the number of false positives generated by the IOD-MAD algorithm is quite low. The end result is an NBM pipeline with relatively low computational demands, which is quite robust, has a limited number of models, and is able to detect 3 different failure types accurately on 5 different wind farms without changes to the configuration of the pipeline.

The overview of the state-of-the-art shows that at the moment a lot of research is done on condition monitoring and failure prediction for wind farms using SCADA data. The NBM methodology is a popular methodological choice for this. Many different configurations (preprocessing, NBM modeling, and analysis of the prediction error) have been tried. Nevertheless, there are still some blind spots that might be interesting for future research. Firstly, a thorough structured analysis of the impact of different preprocessing techniques on the performance of the condition monitoring can be useful. This will give insights into which techniques work well, and give future researchers a basis to start from. This should avoid the situation there is today where often ad hoc decisions are taken without proper motivation why or a clear idea of what the impact will be on the final results. Secondly, a more thorough comparative analysis of different NBM models (e.g. statistical, traditional machine learning, deep learning) might be useful, preferably taking into account the demands/remarks from the industry. Thirdly, most research now focuses on SCADA data with a 10-minute resolution. With more and more data becoming available from wind turbines and improved connectivity, research can be done on data with a higher resolution. Comparative studies of the performance of condition monitoring using 10-minute SCADA data and data with a higher resolution (e.g. 1-minute, 10-second, 1-second) might be interesting. It will give the industry an idea of what the added value is of collecting SCADA data with a higher resolution. Furthermore, it most likely will make it possible to detect events or failures that are short-lived. A thorough comparative analysis of the techniques used for the analysis of the prediction errors would also be useful. Fourthly, these analyses should preferably be done on data from real operational wind farms. Furthermore, it would help the research on this topic a lot, if data from several operational wind farms could be made public. This would make it possible to use these datasets as standards, which would make it easier to compare the performance of different techniques.

*Author contributions.* XC, TV, PJD, JH did the conceptualization; XC developed the methodology, did the data curation and the formal analysis, and wrote the paper; XC, TV, PJD, JH reviewed and edited the paper; JH, AN did the funding acquisition.

*Competing interests.* The authors declare that they have no conflict of interest.

*Acknowledgements.* Xavier Chesterman, Timothy Verstraeten, Pieter-Jan Daems, Ann Nowé, and Jan Helsen received funding from the Flemish Government (AI Research Program). The research presented in this paper is partly financed by the European Union (H2020 PLA-TOON, Pr. No: 872592). The authors would also like to acknowledge FWO for the support through the SBO Robustify project (S006119N), and VLAIO ICON project Supersized 4.0.

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
