# Peer review of "Overview of normal behavior modeling approaches for SCADA-based wind turbine condition monitoring demonstrated on data from operational wind farms"

_Wind Energy Science, 2022_

## Referee Comment (RC2)

The article, titled " Overview of normal behavior modeling approaches for SCADA-based wind turbine condition monitoring demonstrated ondata from operational wind farms" covers a very interesting topic that perfectly fits in the journal's scope.

The structural, linguistic and graphic quality of the publication is very good. The work is clearly structured and the tables, graphs and pictures are easily recognizable and informative.

However, there are a number of major issues that need to be amended or clarified.

Major issues

2.1 Preprocessing techniques.

A list of different techniques for filling/treating missing values is given, but it is needed to give a comprehensive explanation of the benefits/advantages and disadvantages of each one of them. When/why is it recommended to use one or another?

Related to outliers removing, it must be explained how to avoid that abnormal values associated with the failure of interest are also removed.

3.2.2 Selecting healthy training data

It is not clear that healthy data coming from data previous to a failure can be used together with healthy data coming from a data posterior to a failure to construct a NBM. That is because the replacement of a component can significantly affect the (normal) behavior of the WT. Some authors even recommend a "quarantine" of data not to be used after the replacement of a major component such as a gearbox. The WT with the "new" gearbox can have different normal behavior, as in fact it is a different machine. In the paper figures 4 and 5 are too simplistic and do not take the aforementioned comment into account. Explain, how this can be considered or counteracted.

3.3 Normal behavior modelling
In this section, it is written that "The explicit NBM that will mainly be used is the elastic net. It is a simple, transparent, and robust model that can handle large amounts of (correlated) predictors, while at the same time it can work with a limited amount of training data. This corresponds to requirements set by the industry. For this reason, deep learning models, like AEs, are not used in this research."

It is needed more detail about the statement that "industry requires a limited amount of training data". Why this requirement? How limited? The reasoning that deep learning models are not used in this research because of this "requirement of industry" its weak. It is missing an important comparison, as AEs are unsupervised by nature and one of the best fit models for this type of problem. Testing only with elastic net is too simplistic, as deep learning approaches are not compared (only SVR and lightGBM wchich are machine learning approaches). I recommend to compare at least with ANN.

3.4 The anomaly detection procedure

Explain the meaning of variable idio_comp in eq. 5.

Explain the meaning of q in eq. 6.

It should be detailed how variable health_score is computed.

Please, review all equations and explain in detail all variables involved, as they are not clear overall the manuscript.

4.2 Experiment 2: The added value of using lagged predictors
It is said "Reasons for the low added value of the lags can perhaps be an insufficient number of lags, a lack of information on the dynamics in the aggregated SCADA data, or the combination of transient and non-transient behavior. The first hypothesis seems to be unlikely since limited experimentation using more lags showed no clear improvement in performance. The second hypothesis is possible. The third hypothesis would imply that the dynamics of the steady-state and the transient behavior of the turbine are so different they can not be learned by one elastic net model."

I do not agree with the statement. Temperature variables have a slow dynamic, and thus the second hypothesis is not likely to be correct. However, the third hypothesis is very likely, as the WT behavior is highly non-linear, and thus can not be learned by one elastic net model. This is the reason lags did not show improvement. Again, testing only with elastic net is too simplistic, as deep learning approaches are not compared that can really "learn" to highly non-linear dynamics that the WT undergoes.

The experiments given in Sections 4.1, 4.2, 4.3, 4.4, 4.5 are not really tested for early failure prediction, but only tested for how well the NBM predicts the target variable. The title of the paper states that it is an "overview of NBM approaches for SCADA-based wind turbine condition monitoring", therefore this implies testing not only the NBM itself but also the analysis of the NBM prediction error. It is too simplistic to only compare the prediction error on the target variable. It is necessary to include in sections 4.1, 4.2, 4.3, 4.4, 4.5 a study about the prediction error and how many false alarms and good predictions of faults are obtained. For a scientific consideration, a hit rate and an error rate must be given.

In summary, the review poses many questions/hypothesis but fails to focus and answer, through in depth analysis, many of them that are vital. For example:

- Which is the impact of each one of the preprocessing techniques listed (on the early fault detection objective)?
- Which are the benefits/advantages and disadvantages of the different listed techniques for filling/treating missing values?

- Related to outliers removing, how to avoid that abnormal values associated with the failure of interest are also removed?
- Which are the best AI techniques to be used? Here a drawback of the paper is that it is avoiding testing a very important representative part of them (the ones based on deep learning).
- An analysis about how different algorithms for the analysis of the prediction errors affect the final results (number of false alarms and correct alarms on the 5 wind farms over the testing period must be given).

Minor

Reference Catellani et al., 2021 is incorrectly spelled, it should be Castellani.

---

## Author Response (AR1)

**Answers to reviewer 1**

We would like to thank the reviewer for the remarks. In what follows, an answer is given to each remark. The structure of this file is as follows: 1) The question or remark of the reviewer, 2) our reply, 3) location in the annotated manuscript where the relevant changes have been made. It is important to note that for the location of the relevant changes, page and line numbers are used. These numbers match with the location of the changes in the annotated manuscript generated by latexdiff and not necessarily with the location in the revised version of the manuscript.

*The literature review is reasonable but not complete. Useful research has been published on the application of Copula models and also Gaussian Process models in wind turbine normal behaviour modeling. These should be included in the review.*

> The authors of the manuscript acknowledge the fact that research using copula and gaussian process models was missing. The literature overview has been expanded. Copula and Gaussian process based research has been added. The authors would however like to point out that the scope of the literature review has been limited to research that uses normal behavior modelling on temperature signals, meaning that the target should also be a component temperature. A lot of research that uses Copulas and Gaussian processes focuses on the power curve, where the target variable is the active power. This means that this research falls outside the scope of the literature review. The reasoning behind limiting the scope of the literature review is the fact that the topic of the paper focuses strictly on modelling component temperatures, but also that expanding the literature review to also contain the power curve research would inevitably result in a loss of depth of the literature review.

> Changes to the paper:

> - p. 9 lines 262-264: Mentioning of gaussian process and copula based research. Two references have been added.

*I'm not clear why the data filtering reduced the normall 10 minute data to hourly. Better results may have been obtained using the 10 minute data directly (after error and gap removal). If noise reduction was the main reason this should be demonstrated statistically.*

> The reasoning behind reducing the resolution of the data to hour level (from 10 minute level) through aggregation is threefold. Firstly, component temperatures tend to change relatively slowly over time, which makes them more suitable for the detection of relatively long-term evolutions. This means that for this case the difference between the 10-minute or 1-hour resolution is marginal or non-existent, something that became clear during experimentation. Secondly, aggregating the data results in noise reduction. This was shown in Turnbull (2021). Thirdly, it reduces the computational burden of the methodology considerably, which is one of the main focus points of this manuscript. The authors acknowledge nevertheless that there are cases where this trade-off should not be done. This is the case for failures that form very fast over time, or for signals that exhibit damage patterns that are very short-lived like in vibration analysis. This explanation has been added to the paper.

> Changes to the paper:

- p. 13 lines 352-356: Explanation when the aggregation to 1-hour level can be used and when not has been added.

*The mathematics of the modelling and error prediction should be presented in more detail. The terms in equation 2 should be explained.*

The authors of the manuscript acknowledge the fact that the equations were not sufficiently explained. A more thorough explanation of the equations has been added to the revised manuscript.

Changes to the paper:

- p. 19 around lines 465-469 equation 3 (equation 2 has become equation 3): An explanation/definition of the different parts of the equation has been added.
- Furthermore, even though the reviewer did not explicitly asked for it, more elaborate explanations have been added to the other equations.

*Cooling transients are a major cause of modelling difficulty. However they do not represent an operational turbine and I'm surprised such data was not excluded from the analysis by using status or power data from the SCADA,*

The power downs of the turbine have been removed from the data using the active power as a guideline. The transient behavior has also been removed to a large extent from the data by subtracting the fleet median from the signals. Figure 2 shows that transient behavior that is common to the fleet is captured by the common component (fleet median), and that the idiosyncratic component is free of it. This type of transient behavior is caused for example by cool-downs due to lack of wind, … What remains of transient behavior is unique to the turbine. This can for example be caused by maintenance of the wind turbine. These events are however much more rare. So most transient behavior has been removed by subtracting the fleet median from the signals. Furthermore, the data has been aggregated to hour-level, which further reduces the remaining transient behavior.

Changes to the paper:

- No changes to the paper required.

**Answers to reviewer 2**

We would like to thank the reviewer for the remarks. In what follows, an answer is given to each remark. The structure of this file is as follows: 1) The question or remark of the reviewer, 2) our reply, 3) location in the annotated manuscript where the relevant changes have been made. It is important to note that for the location of the relevant changes, page and line numbers are used. These numbers match with the location of the changes in the annotated manuscript generated by latexdiff and not necessarily with the location in the revised version of the manuscript.

**Major issues**

**2.1 Preprocessing techniques.**

*A list of different techniques for filling/treating missing values is given, but it is needed to give a comprehensive explanation of the benefits/advantages and disadvantages of each one of them. When/why is it recommended to use one or another? Related to outliers removing, it must be explained how to avoid that abnormal values associated with the failure of interest are also removed.*

The authors acknowledge that the part on the preprocessing techniques mainly sums up the different techniques. We have added some extra information (including a review paper) to the paper that explains what potential problems there might arise with certain technique. Still, we do believe that an in-depth analysis of the different preprocessing techniques falls outside the scope of the paper, as the main focus remains on the NBM and anomaly detection techniques. Furthermore, some extra information has been added to the paper concerning the outlier handling.

Changes to the paper:

- p. 4-5 lines 95-126: Extra information on imputation has been added to the paper.
- p. 5 around lines 132-138: Some extra information on outlier detection has been added to the paper.

**3.2.2 Selecting healthy training data**

*It is not clear that healthy data coming from data previous to a failure can be used together with healthy data coming from a data posterior to a failure to construct a NBM. That is because the replacement of a component can significantly affect the (normal) behavior of the WT. Some authors even recommend a "quarantine" of data not to be used after the replacement of a major component such as a gearbox. The WT with the "new" gearbox can have different normal behavior, as in fact it is a different machine. In the paper figures 4 and 5 are too simplistic and do not take the aforementioned comment into account. Explain, how this can be considered or counteracted.*

The authors acknowledge the fact that a new generator or gearbox bearing can behave differently compared to the previous one. However, data analysis showed that the differences in temperatures are quite small. Experiments have been done where the models were retrained after each replacement, for each turbine separately. However, there was no clear difference in performance. This indicates that, even though it is likely that there are some differences between the old and new bearings, the differences are so small that they do not jump out compared to the noise that remains after modelling. The advantages in terms of computational complexity, training time and required amount of training data has made us decide to go with a single training phase. This information has been added to the revised manuscript. We have not included the results of these experiments to keep the number of pages in the manuscript limited.

Furthermore, the authors acknowledge the importance of quarantining data that comes from just after a replacement. This is also done in our research. Data that follows a replacement by less than a month is not used for training (see Section 3.2.2). This to avoid that the training data gets polluted with upstart and testing behavior. We acknowledge that figure 4 is too simplistic. Figure 4 shows which data is considered

healthy (and suitable for training) and which data is considered unhealthy. However, the figure contained an error. It did not show the 1 month quarantining of the data. The figure has been corrected in the revised version of the manuscript.

Changes to the paper:

- p. 16 line 401: Figure 4 has been corrected.

**3.3 Normal behavior modelling**

*In this section, it is written that "The explicit NBM that will mainly be used is the elastic net. It is a simple, transparent, and robust model that can handle large amounts of (correlated) predictors, while at the same time it can work with a limited amount of training data. This corresponds to requirements set by the industry. For this reason, deep learning models, like AEs, are not used in this research." It is needed more detail about the statement that "industry requires a limited amount of training data". Why this requirement? How limited? The reasoning that deep learning models are not used in this research because of this "requirement of industry" its weak. It is missing an important comparison, as AEs are unsupervised by nature and one of the best fit models for this type of problem. Testing only with elastic net is too simplistic, as deep learning approaches are not compared (only SVR and lightGBM wchich are machine learning approaches). I recommend to compare at least with ANN.*

The authors of the manuscript acknowledge that the statement "*The explicit NBM that will mainly be used is the elastic net. It is a simple, transparent, and robust model that can handle large amounts of (correlated) predictors, while at the same time it can work with a limited amount of training data. This corresponds to requirements set by the industry. For this reason, deep learning models, like AEs, are not used in this research.*" is quite strongly worded (more strongly than was actually intended). We did not mean that AE models are not used in the industry. We are aware of the fact that they are used. What it should have been is that the requirements were set by the industrial partners with whom we collaborated. We do however believe that the requirements set by the industrial partners correspond in general to the requirements of the whole wind turbine operator industry.

Based on discussions with the industrial partners, a scope has been defined for this research. The scope was defined as developing a methodology that can handle situations in which there is a constraint on the amount of training data (at most a couple of months of training data per turbine, but preferably less) (using 10-minute SCADA data). Other demands were that the models are fast to train, and do not require a lot of maintenance (this was never precisely quantified). The idea behind these requirements is that when a new wind farm is added to the system, the condition monitoring methodology should be operational in a short time. These requirements explain why the models were trained on a limited amount of training data (6 months), why experiments have been done with only 2 months of training data, why the elastic-net vs. the rest analysis was done (elastic-net trains much faster), …, and it also explains why deep learning was not taken into account.

We acknowledge that this was not made sufficiently clear in the manuscript. The clarifications have been added. Furthermore, the revised version of the paper is expanded with results from multi-layer perceptrons. The results show/confirm the results presented in the original paper. In some cases these models perform (marginally)

better than the elastic-net model. In other cases they perform worse than the elastic-net. The results are more or less in line with the results from the light GBM and the conclusions derived from it. An explanation for the fact that the more complex models only perform marginally better in some cases can be found in the preprocessing. By subtracting the fleet median from the SCADA signals, most transient behavior has been removed (it is "modelled" out by the fleet median which can be considered an implicit NBM). This "simplifies" the problem substantially, which means that the better modelling qualities from the more complex models do not really show.

Changes to paper:

- p.18 lines 446-453: The expression "*The explicit NBM that will mainly be used is the elastic net. It is a simple, transparent, and robust model that can handle large amounts of (correlated) predictors, while at the same time it can work with a limited amount of training data. This corresponds to requirements set by the industry. For this reason, deep learning models, like AEs, are not used in this research.*" has been clarified.
- Results for multi-layer perceptrons have been added to the paper (see Figure 12 p. 27, Figure 20 p.33, Figure 21 p.35, Figure 22 p.35, Figure 23 p.36 and Figure 24 p.37 + accompanying explanations).

**3.4 The anomaly detection procedure**

*Explain the meaning of variable idio_comp in eq. 5. Explain the meaning of q in eq. 6. It should be detailed how variable health_score is computed. Please, review all equations and explain in detail all variables involved, as they are not clear overall the manuscript.*

The authors of the manuscript acknowledge that the explanation of the equations was lacking. For this reason more detailed information on the equations has been added. An equation explaining the calculation of the health score has also been added.

Changes to paper:

- p. 21 equation 6 (formerly equation 5): More detailed explanation of the equation has been added to the paper.
- p. 21 equation 7 (formerly equation 6): More detailed explanation of the equation has been added to the paper.
- More detailed explanations have been added to the other equations in the manuscript.

**4.2 Experiment 2: The added value of using lagged predictors**

*It is said "Reasons for the low added value of the lags can perhaps be an insufficient number of lags, a lack of information on the dynamics in the aggregated SCADA data, or the combination of transient and non-transient behavior. The first hypothesis seems to be unlikely since limited experimentation using more lags showed no clear improvement in performance. The second hypothesis is possible. The third hypothesis would imply that the dynamics of the steady-state and the transient behavior of the turbine are so different they can not be learned*

*by one elastic net model." I do not agree with the statement. Temperature variables have a slow dynamic, and thus the second hypothesis is not likely to be correct. However, the third hypothesis is very likely, as the WT behavior is highly non-linear, and thus can not be learned by one elastic net model. This is the reason lags did not show improvement. Again, testing only with elastic net is too simplistic, as deep learning approaches are not compared that can really "learn" to highly nonlinear dynamics that the WT undergoes.*

The authors would like to thank the referee for these remarks. However, we would like to point out that in the paper no decision was taken on which hypothesis (the 2$^{nd}$ or 3th) is the correct one. The only thing that has been stated is that the 1th hypothesis is unlikely. Further research is required to determine which of the two remaining hypothesis are correct. This will be postponed to a future paper.

Furthermore, we would like to point out that the non-transient behavior problem has largely been resolved by subtracting the fleet median from the signals. By doing this, transient behavior that is common to the whole fleet, which means it is caused by environmental conditions, is removed. This can be seen in Figure 2 and 3, where the idiosyncratic component exhibits much less variance. This is due to the fact that common cool-downs are captured by the common component. The transient behavior that remains in the idiosyncratic component are for example cool-downs that are specific to a turbine. These can be the result of a turbine that is stopped for maintenance. However, these types of events are much more rare than the common cool-downs. This means that when the idiosyncratic component is used as input to other explicit NBM models like the elastic net, the problem to be solved has been simplified substantially.

It is in our opinion not so clear that the second hypothesis is invalid. If we would use the original SCADA data signals as input, the hypothesis would indeed be unlikely. However, when the fleet median has been subtracted from the SCADA data signals this is not so clear anymore. Let us for the sake of the argument assume that all wind turbines have exactly the same signal values (for example all have the same generator bearing temperature). This implies that the fleet median would be exactly the same as the generator bearing temperatures of all the wind turbines. This means that if the fleet median is subtracted from these temperatures, the result (the idiosyncratic component) equals to 0, no matter whether the turbines are in steady-state or in transient behavior (we ignore here the case in which a single turbine is transient, e.g., due to maintenance). This means that there is no autocorrelation left in the signal, which means that adding lagged values to the model is useless. In reality of course the generator bearing temperatures of the different turbines are not exactly the same. There are some small deviations, causing the idiosyncratic component of a turbine to be sometimes slightly above or below zero. This means that the idiosyncratic component most likely contains at least some autocorrelation. However, this is substantially less than in the original signal. Furthermore, adding the lags of the different predictors massively increases the dimensionality of the problem, which can impact the performance of the algorithms. These two phenomenon together can explain, in our opinion, why adding lagged values might only have a limited impact on the performance results. In conclusion, in our opinion it is not clear whether the second or third hypothesis is the most likely. This can be the topic of future research.

We do however acknowledge that this reasoning has not been sufficiently explained in the paper. For this reason it has been added. Furthermore, we expanded experiment 2 so that the revised paper contains also results for multi-layer perceptrons.

Changes to paper

- p.25 – 28 lines 594 - 632: Results for multi-layer perceptrons have been added to the paper + the discussion of the different hypothesis has been expanded.

*The experiments given in Sections 4.1, 4.2, 4.3, 4.4, 4.5 are not really tested for early failure prediction, but only tested for how well the NBM predicts the target variable. The title of the paper states that it is an "overview of NBM approaches for SCADA-based wind turbine condition monitoring", therefore this implies testing not only the NBM itself but also the analysis of the NBM prediction error. It is too simplistic to only compare the prediction error on the target variable. It is necessary to include in sections 4.1, 4.2, 4.3, 4.4, 4.5 a study about the prediction error and how many false alarms and good predictions of faults are obtained. For a scientific consideration, a hit rate and an error rate must be given.*

We agree that condition monitoring is not only the normal behavior modelling, but also the detection of the anomalies. However, since it is not possible to check all the different combinations of the NBM modelling and the anomaly detection techniques, we decided to split the research in two. Sections 4.1-4.5 focus on the normal behavior modelling, in which attention goes to how well the model can predict the healthy test data. We started from a benchmark in which only the fleet median is used. Then, we applied the elastic net on top of it. We saw an improvement in the performance. Next, we tested several strategies, such as adding lags, more complex machine learning models, … The conclusion was that the base elastic net model predicts relatively well the healthy test data, and that the different modifications (and other ML models) have no or only limited added value.

Once this was established we used the simplest well performing model (i.e. the base elastic net model), and used this in the experiment on the anomaly detection techniques (see Section 4.6). The idea behind this decision was that training time, complexity, … play an important role. For this reason it seems to us that the decision to use the base elastic net model as a basis to analyze the performance of the anomaly detection techniques is reasonable.

However, we do acknowledge that simply mentioning the RMSE of the prediction error on the healthy test data (as done in experiments 1-5) is insufficient. For this reason we constructed a new metric that analyses the difference between the median prediction error for the unhealthy and healthy data. A good normal behavior model for the cases that are analyzed in the manuscript has a large positive value for this metric. This is because of the fact that the damaged components that are related to the failures tend to have higher temperatures than under normal/healthy conditions. The larger this value is, the better the model is for making a distinction between healthy and unhealthy behavior, and the more useful it is for anomaly detection. The results for the new metric are added to each experiment (except experiments 1 and 6) in the revised manuscript.

Changes to paper:

- p. 18 lines 432-438: Description of the second metric has been added.
- Figures that show the results for the second metric have been added to the paper (p.27 Figure 12, p.31 Figure 16, p.33 Figure 20, p.37 Figure 24) + explanation of the results shown in the figures has been added.

**Minor issues**

Reference Catellani et al., 2021 is incorrectly spelled, it should be Castellani.

This error is corrected in the revised paper.

Changes to paper:

p.46 line 898: Correction done in references section